# Can working in agriculture have a favorable effect on depressive symptoms? Life satisfaction as a mediator

Feiying He[1‡], Xiaoying Li[2‡], Xiangchun Xu[3‡], Shulin Bao[2], Yanwu Chen[4,5], Hualin Liu[4,5*], Yuan Yao Chen ORCID [4,5*]

1 School of Health Management, Southern Medical University, Guangzhou, China, 2 School of Basic Medicine, Southern Medical University, Guangzhou, China, 3 Nanfang Hospital, Southern Medical University, Guangzhou, China, 4 Research & Development Division, Perfect Life & Health Institute, Zhongshan, China, 5 Research & Development Division, Perfect (Guangdong) Co., Ltd., Zhongshan, China

‡ FH, XL, and XX contributed equally to this work and share first authorship.
* yuanyaochen0401@163.com (YYC); lhl371@126.com (HL)

## Abstract

### Purpose

Several studies have explored the relationship between various aspects of work and the onset of depressive symptoms. However, there is a lack of research focusing on the association between job types and depressive symptoms. This study aims to investigate the impact of agricultural work on depressive symptoms and whether life satisfaction mediates this relationship.

### Methods

Data were obtained from the 2015 China Health and Retirement Longitudinal Study (CHARLS) (n = 6856). Participants were categorized based on whether they were engaged in agricultural or non-agricultural work and further classified as self-employed or employed. Depressive symptoms and life satisfaction were assessed using the CES-D and SWLS scales. Logistic regression analysis was used to examine associations, and Baron and Kenny's mediation test and the Sobel test were used to assess the mediating effect of life satisfaction.

### Results

Engaging in agricultural work was positively associated with increased depressive symptoms scores (B = 3.437, p < 0.001), indicating that agricultural work exacerbates depressive symptoms. This effect was partially mediated by life satisfaction.

**Data availability statement:** All relevant data can be found at the following website: http://CHARLS.pku.edu.cn/en. We conducted the study just through downloading the three questionaires called "Demographic_Background," "Health_Status_and_Functioning" and "Work_Retirement_and_Pension" from CHARLS 2015, all of which can be found at: https://charls.charlsdata.com/pages/data/111/en.html.

**Funding:** This research is supported by Key Laboratory of Philosophy and Social Sciences of Colleges and Universities in Guangdong Province (2023WSYS005, 2023WSYS007), the 13th Five-Year Plan of Guangdong Province for Philosophy and Social Sciences (GD20XGL42). There is no funders participating in the study or in the process of study.

**Competing interests:** The authors have declared that no competing interests exist.

## Conclusions

Self-employed agricultural workers are a high-risk group for depressive symptoms. Additionally, life satisfaction plays a mediating role between type of job and depressive symptoms. Public health recommendations aimed at improving or mitigating depressive symptoms among agricultural workers could focus on enhancing life satisfaction to promote healthier psychological status.

## Introduction

Depression is a common mental illness and has now become the biggest health burden worldwide [1,2]. According to the World Health Organization, about 5% of adults suffer from depression worldwide, affecting 280 million people, of whom more than 54 million people in China suffer from depression [3]. The number of people suffering from depression is still on the rise with each passing year. Given the impact of depression on individuals and society, there is a great need for in-depth research on depression.

Research has shown that work is closely related to the development of depression [4], which can be subdivided into work pressure [5], work hours [6,7], work environment [8,9], work support [10], subjective social status [11], and so on, but a lack of research focusing on the association between job type of agricultural and non-agricultural and depression. Among the mainstream treatments for depression, there is a therapy called Horticultural Therapy [12,13], which suggests that gardening and being close to nature can help people with depression to recover. While empirical research has demonstrated a significantly elevated prevalence of depressive symptoms among middle-aged and elderly populations residing in rural China compared to their urban counterparts [14]. This epidemiological phenomenon stands in stark contrast to the theoretical postulates of Horticultural Therapy, which advocate the mental health benefits derived from nature immersion. The findings reveal a complex mechanistic relationship between agrarian occupational environments and psychological well-being, indicating that mere proximity to natural elements fails to comprehensively elucidate the etiological pathways through which agricultural labor influences depressive states. Life satisfaction is an important indicator of subjective well-being, which can reflect an individual's cognitive evaluation of quality of life. It has been proved that there is an association between life satisfaction and depression in middle-aged and elderly people [15,16], which can be used as an important indicator of mental health and quality of life of middle-aged and elderly people. And some other factors of work have been shown to be associated with life satisfaction [17–19]. Therefore, in this study, it is necessary to investigate whether the type of jobs has a correlation with life satisfaction

This study will first examine the association between different types of work and depressive symptoms. Further, it will analyze population subgroups based on work state—self-employed versus employed—in order to provide more targeted public health recommendations. Additionally, life satisfaction will be incorporated into the

model as a mediating variable to explore whether job types influence depressive symptoms through the pathway of life satisfaction. It was hypothesized that agricultural work could reduce depressive symptoms by increasing subjects' life satisfaction. The relationship between farming, life satisfaction, and depressive symptoms was empirically investigated as a way to provide a theoretical basis for national mental health promotion.

## Method

### Research design

The data comes from the 2015 China Health and Aging Tracking Survey (CHARLS), a large-scale social survey project jointly initiated and undertaken by the National Development Research Institute and the School of Management at Peking University. The database aims to collect tracking survey data on the health, economic and social conditions of China's middle-aged and elderly population aged 45 and above, providing important data support for the study of China's aging problem.

The survey in the CHARLS database covers a wide range of aspects such as the health states, lifestyles, economic states, social security, family structures and mental health of middle-aged and elderly people. The survey adopts a multi-stage random sampling method to ensure the representativeness of the sample. The questionnaire was rigorously designed, and internationally recognized indicators and measurement methods were used to ensure the comparability of the data. All subjects signed an informed consent form and were approved by the Ethics Committee of Peking University (IRB00001052–11015).

### Ethics approval

All respondents gave informed written consent prior to the interview, and ethical approval for the collection of human subject data was obtained, which was updated annually at the Peking University Institutional Review Board (IRB00001052–11015). No additional ethical approval is required for approved data users. All methods of this study were performed in accordance with the guidelines of the Declaration of Helsinki.

### Study population

In the present study, participants were from CHARLS (2015) and the criteria for inclusion of subjects at baseline were subjects who contained a record of type of jobs, assessment of depressive states, assessment of life satisfaction, age, gender, type of residence, work states, marital states, socially engaged activity score, smoking states, drinking states, sleep duration, and prior work states in CHARLS 2015. Of these, missing values for job type records (n = 23), missing values for depressive symptoms (n = 1297) missing values for life satisfaction (n = 1436), and missing values for other covariates (n = 11270) were excluded, and 6856 subjects were ultimately enrolled in the study (Fig 1).

### Measurement tools

**Job type and work state.** In this study, the population was grouped by type of jobs and work states according to the questionnaire questions of CHARLS 2015 WORK, RETIREMENT AND PENSION. The types of jobs was categorized into those engaged in agricultural work and those engaged in non-agricultural work, and the work states were categorized into self-employed and employed. The exact division steps can be seen in the following figure (Fig 2).

### Depressive symptoms

Depression state was measured using the 10-item Center for Epidemiological Studies Depression Scale (CES-D), which is now a widely used and proven valid and reliable common indicator of depression [20,21]. Previous studies have shown that this assessment is a sensitive, responsive, valid, and reliable tool for identifying and tracking depressive symptoms

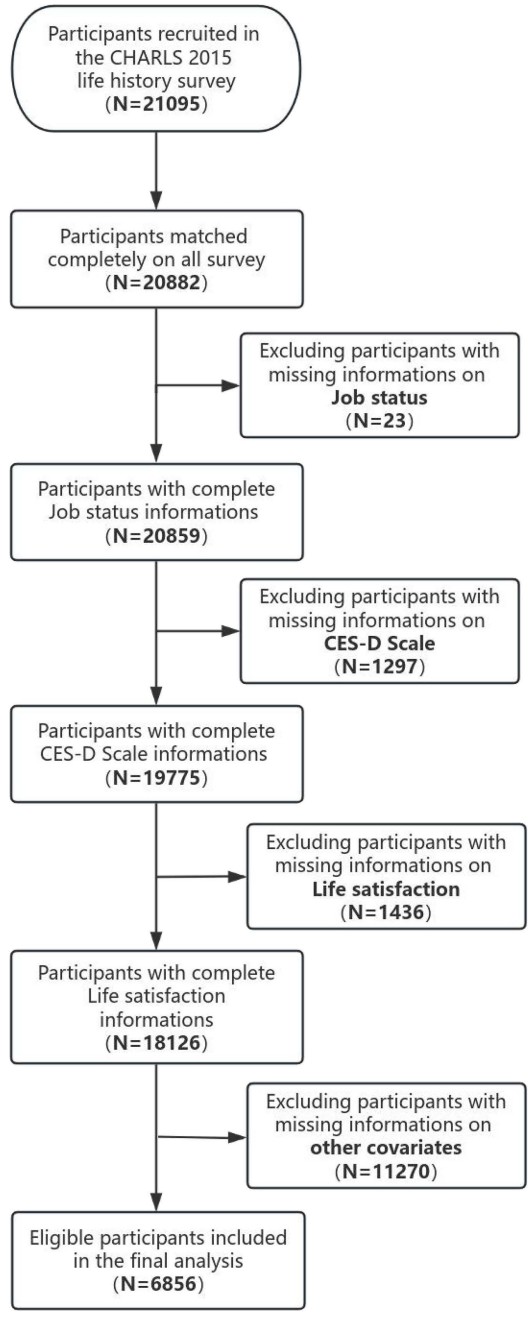

**Fig 1. Flowchart ofparticipants selection.**

in Chinese adults. The CES-D assesses depression through 10 dimensions, including 8 negative emotions (e.g., I feel depressed) and 2 positive emotions (e.g., I am hopeful about the future). Participants were asked to assess how often they felt each of the ten emotions in the past week. Responses to each dimension were scored and coded on a 4-point scale ranging from 0 "little or no time (< 1 day)" to 3 "most or all of the time (5 - 7 days)". The final CES – D score was calculated by combining the ten dimensions, ranging from 0–30, and was corrected for two positive emotions. The higher the CES – D score, the more severe the depression.

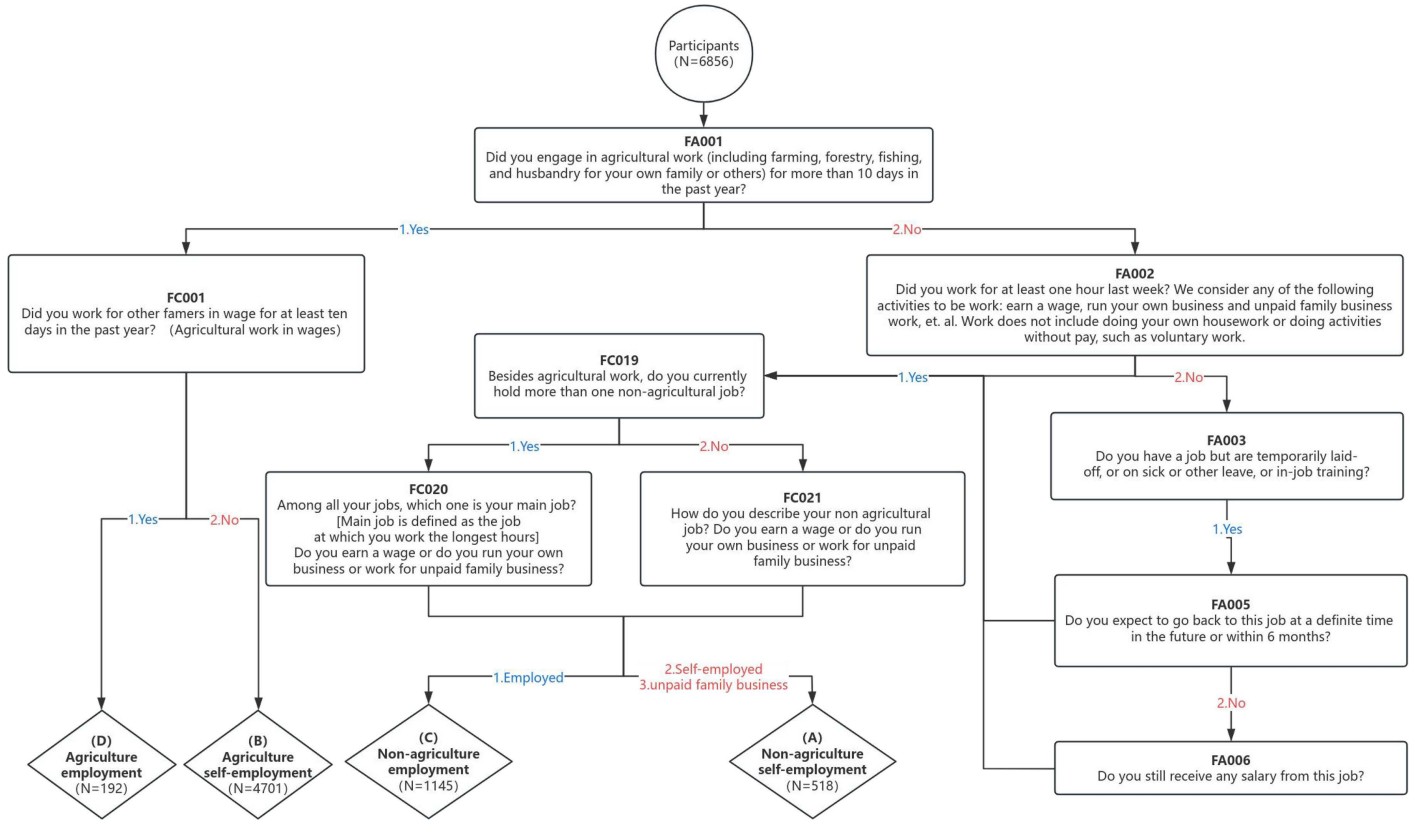

**Fig 2. Subgroup division.**

## Life satisfaction

Life satisfaction was assessed at baseline by asking participants, "How satisfied are you with your life in general?". "Life satisfaction was assessed at baseline. This question correlates strongly with the validated 5-item Satisfaction with Life Scale (SWLS) [22] and has been used extensively in previous studies. The five response options are: not at all satisfied, not too satisfied, relatively satisfied, very satisfied, and extremely satisfied. The scores range from 0~4, with higher scores indicating higher life satisfaction.

## Selection of covariates

Based on previous studies, we selected a series of covariates, including demographic characteristics such as age [23–25], gender [26,27], type of residential address [28], and marital states [29,30], as well as factors that have been validated by research to be highly correlated with depressive states, such as social activity participation [31], smoking states [32–34], drinking states [35]. These covariates were obtained from CHARLS: gender was categorized as 0 (female) and 1 (male), type of residential address was categorized as 0 (urban) and 1 (rural), and marital states were categorized as 0 (unmarried) and 1 (married), and social activity participation was assessed by using the question DA056 in CHARLS 2015, which asked subjects "In the past month, did you engage in the following Social activities" was asked in question DA056 of CHARLS 2015, and scores were assigned according to the social activities selected, with each item accumulating one point, ranging from 0 to 11 points, with higher scores indicating higher participation in social activities. Smoking states were categorized as 0 (non-smokers), 1 (smokers); drinking states were rated using a continuous value score,

based on question DA067 in the self-answered CHARLS 2015, with scores ranging from 0~2, with higher scores indicating more frequent drinking. Prior work states data were derived from the ZF1 loading variable of CHARLS 2015, categorized as 0 (not working at the time of the last round of the survey), 1 (working in non-farming at the time of the last round of the survey), 2 (working in agriculture at the time of the last round of the survey), and 3 (working in both agriculture and non-farming at the time of the last round of the survey).

## Methods of statistical analysis

As a preliminary analysis, a presentation of baseline data was conducted. Correlation analyses were then conducted to reveal associations between job type, life satisfaction, and depressive states, as well as with covariates. After clarifying the correlations between variables, we dichotomized depression states for logistic regression and divided the population by job type and job states for subgroup regression analysis. We then conducted mediation effects analyses and performed sensitivity tests. We used Baron and Kenny's mediation test [36,37] to access the mediating role of life satisfaction. Sobel's test [37,38] was also used to verify the sensitivity of the mediating effect. Finally, we propose a mediated effects model. The above statistical analysis is based on R-4.3.1.

## Result

### Sample characteristics before matching

The socio-demographic characteristics of the study subjects are shown in Table 1. A total of 6856 subjects were included in this study, of whom 1663 were engaged in non-agricultural work and 5193 in agricultural work. A significantly higher proportion of subjects were females (48.8%) working in agriculture than those working in non-agriculture (38.7%), and the mean age of agricultural workers (68.2 years) was higher than that of non-agricultural workers (64.3 years). Most of the subjects working in agriculture had rural residential addresses (89.9%) and self-employed working states (90.5%), while most of the subjects working in non-agriculture had urban residential addresses (57.3%) and employed working states (68.9%). There was no significant difference in the marital states of the subjects.

The mean CES-D scores of agricultural workers were significantly higher than those of non-agricultural workers (agricultural: 8.37, non-agricultural: 6.17). In addition to this, there were significant differences between the two groups of subjects in life satisfaction score (p = 0.0108), social activity participation score (p < 0.001), smoking states (p = 0.00442), drinking states (p < 0.001), sleep duration (p = 0.0263), and prior work states (p < 0.001).

### Sample characteristics after matching

After propensity matching scores, the socio-demographic characteristics of the study subjects after matching those who worked in agriculture and those who did not are shown in Table 2. 1,663 people worked in non-agricultural jobs and 1,663 people worked in agriculture after matching. Agricultural workers had a higher proportion of females (69.3% vs. 38.7%, p < 0.001) and a higher mean age (76.1 years vs. 64.3 years, p < 0.001) compared to non-agricultural workers. All agricultural workers lived in rural areas and were mostly self-employed (93.1%), while most non-agricultural workers lived in urban areas (57.3%) and were employed (68.9%). There was no significant difference in the marital states of the subjects.

The mean CES-D scores of agricultural workers were significantly higher than those of non-agricultural workers (agricultural: 9.79, non-agricultural: 6.17).

### Relevance analysis

In order to understand the relationship between the variables and the mechanism of action, the Spearman's correlation analysis of the variables is shown in Fig 3. Overall, age was positively correlated with type of job

**Table 1. Baseline before matching.**

| | Non-agriculture (N = 1663) | Agriculture (N = 5193) | p-value |
|---|---|---|---|
| **Gender** | | | |
| Female | 643 (38.7%) | 2532 (48.8%) | <0.001 |
| Male | 1020 (61.3%) | 2661 (51.2%) | |
| **Age** | | | |
| Mean (SD) | 64.3 (6.85) | 68.2 (8.30) | <0.001 |
| Median [Min, Max] | 63.0 [28.0, 91.0] | 68.0 [28.0, 101] | |
| **Address_type** | | | |
| City | 953 (57.3%) | 526 (10.1%) | <0.001 |
| Village | 710 (42.7%) | 4667 (89.9%) | |
| **Work_state** | | | |
| Self-employed | 518 (31.1%) | 4701 (90.5%) | <0.001 |
| Employed | 1145 (68.9%) | 492 (9.5%) | |
| **Martial_states** | | | |
| Unmarried | 119 (7.2%) | 448 (8.6%) | 0.0651 |
| Married | 1544 (92.8%) | 4745 (91.4%) | |
| **CES_D_score** | | | |
| Mean (SD) | 6.17 (5.30) | 8.37 (6.38) | <0.001 |
| Median [Min, Max] | 5.00 [0, 29.0] | 7.00 [0, 30.0] | |
| **Life_satisfaction_score** | | | |
| Mean (SD) | 2.44 (0.735) | 2.39 (0.774) | 0.0108 |
| Median [Min, Max] | 2.00 [0, 4.00] | 2.00 [0, 4.00] | |
| **Social_activity_score** | | | |
| Mean (SD) | 1.66 (1.19) | 1.31 (0.795) | <0.001 |
| Median [Min, Max] | 1.00 [1.00, 9.00] | 1.00 [0, 8.00] | |
| **Smoke_state** | | | |
| Nonsmoker | 1076 (64.7%) | 3557 (68.5%) | 0.00442 |
| Smoker | 587 (35.3%) | 1636 (31.5%) | |
| **Drinking_state** | | | |
| Mean (SD) | 0.823 (0.926) | 0.694 (0.907) | <0.001 |
| Median [Min, Max] | 0 [0, 2.00] | 0 [0, 2.00] | |
| **Sleep_time** | | | |
| Mean (SD) | 6.50 (1.56) | 6.40 (1.95) | 0.0263 |
| Median [Min, Max] | 7.00 [0, 12.0] | 6.50 [0, 15.0] | |
| **Previous_work_type** | | | |
| Not Work | 219 (13.2%) | 438 (8.4%) | <0.001 |
| Non-agricultural Work | 1047 (63.0%) | 261 (5.0%) | |
| Agricultural Work | 218 (13.1%) | 3687 (71.0%) | |
| Both | 179 (10.8%) | 807 (15.5%) | |

(r = 0.68, p < 0.05), type of residence (r = 0.49, p < 0.05) and social activity participation score (r = 0.25, p < 0.05) and with marital states (r = −0.18, p < 0.05), CES-D score (r = −0.22, p < 0.05) and prior work states (r = −0.49, p < 0.05), while there was no correlation with gender, life satisfaction score, smoking states, drinking states and sleep duration.

**Table 2. Baseline after matching.**

| | Non-agriculture (N = 1663) | Agriculture (N = 1663) | P-value |
|---|---|---|---|
| **Gender** | | | |
| Female | 643 (38.7%) | 1152 (69.3%) | <0.001 |
| Male | 1020 (61.3%) | 511 (30.7%) | |
| **Age** | | | |
| Mean (SD) | 64.3 (6.85) | 76.1 (5.98) | <0.001 |
| Median [Min, Max] | 63.0 [28.0, 91.0] | 76.0 [65.0, 101] | |
| **Address_type** | | | |
| City | 953 (57.3%) | 0 (0%) | <0.001 |
| Village | 710 (42.7%) | 1663 (100%) | |
| **Work_state** | | | |
| Self-employed | 518 (31.1%) | 1548 (93.1%) | <0.001 |
| Employed | 1145 (68.9%) | 115 (6.9%) | |
| **Martial_status** | | | |
| Unmarried | 119 (7.2%) | 200 (12.0%) | <0.001 |
| Married | 1544 (92.8%) | 1463 (88.0%) | |
| **CES_D_score** | | | |
| Mean (SD) | 6.17 (5.30) | 9.79 (6.86) | <0.001 |
| Median [Min, Max] | 5.00 [0, 29.0] | 8.00 [0, 30.0] | |
| **Life_satisfaction_score** | | | |
| Mean (SD) | 2.44 (0.735) | 2.37 (0.770) | 0.00708 |
| Median [Min, Max] | 2.00 [0, 4.00] | 2.00 [0, 4.00] | |
| **Social_activity_score** | | | |
| Mean (SD) | 1.66 (1.19) | 1.08 (0.458) | <0.001 |
| Median [Min, Max] | 1.00 [1.00, 9.00] | 1.00 [0, 5.00] | |
| **Smoke_state** | | | |
| Nonsmoker | 1076 (64.7%) | 1324 (79.6%) | <0.001 |
| Smoker | 587 (35.3%) | 339 (20.4%) | |
| **Drinking_state** | | | |
| Mean (SD) | 0.823 (0.926) | 0.515 (0.840) | <0.001 |
| Median [Min, Max] | 0 [0, 2.00] | 0 [0, 2.00] | |
| **Sleep_time** | | | |
| Mean (SD) | 6.50 (1.56) | 6.10 (2.20) | <0.001 |
| Median [Min, Max] | 7.00 [0, 12.0] | 6.00 [0, 15.0] | |
| **Previous_work_type** | | | |
| Not Work | 219 (13.2%) | 206 (12.4%) | <0.001 |
| Non-agricultural Work | 1047 (63.0%) | 28 (1.7%) | |
| Agricultural Work | 218 (13.1%) | 1343 (80.8%) | |
| Both | 179 (10.8%) | 86 (5.2%) | |

## Logistic regression

Next, in order to explain the relationship between the variables and the depressive state and to find the risk factors for the depressive state, a Logistic multi-factorial regression model was developed (Table 3).Logistic regression analysis revealed that being male (OR = 0.52, 95% CI 0.37 to 0.72, p < 0.001), married (OR = 0.47, 95% CI 0.34 to 0.64, p < 0.001), and employed (OR = 0.56, 95% CI 0.40 to 0.79, p < 0.001) were significantly protective against depressive states. Additionally,

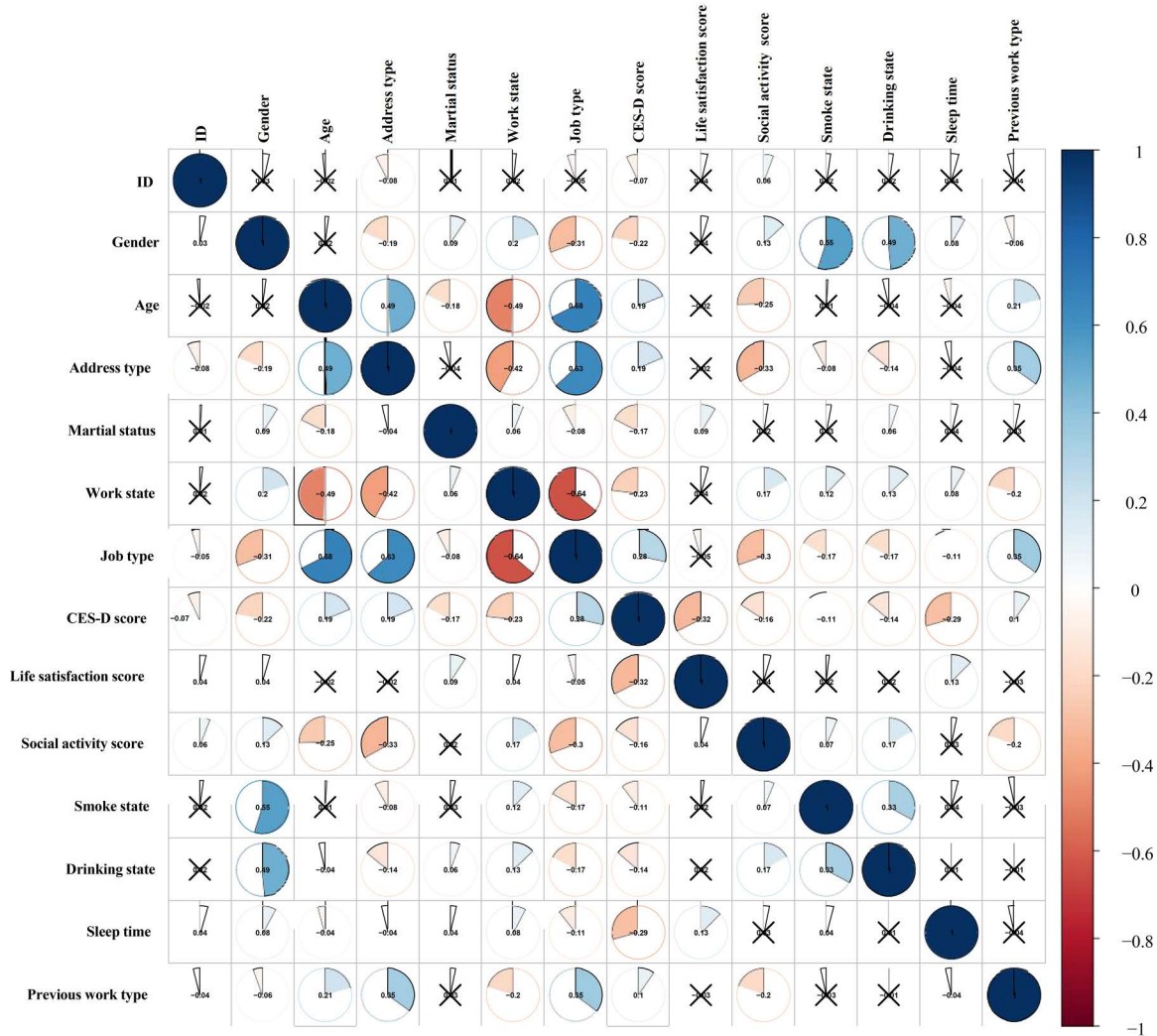

The Spearman correlation analysis shows the correlations between age, job type, address type, social activity participation, marital status, CES-D scores, and prior work type.

**Fig 3. Correlation analysis-relationships between variables.**

life satisfaction (OR = 0.41, 95% CI 0.36 to 0.48, p < 0.001), social activity participation (OR = 0.60, 95% CI 0.47 to 0.76, p < 0.001), and sleep duration (OR = 0.77, 95% CI 0.73 to 0.81, p < 0.001) had significant protective effects against depressive states.

## Subgroup regression analysis

In order to explore in more depth the differences in risk factors for depressive states across populations, we divided the population into six subgroups according to two categorization criteria – job type and job states.

## Job type subgroups

The subjects were divided into two subgroups, agricultural and non-agricultural, by type of jobs, and the results of the logistic regression are shown in Fig 4. Among smokers, subjects working in non-agricultural jobs compared to those working in agricultural jobs had an approximately 1.80-fold increased risk of developing a depressive state when compared

**Table 3. Logistic regression.**

| Dependent: CES-D score | | Low (N = 2858) | High (N = 468) | OR (multivariable) |
|---|---|---|---|---|
| **Job type** | Non-agriculture | 1554 (54.4%) | 109 (23.3%) | |
| | Agriculture | 1304 (74%) | 359 (76.7%) | 1.07 (0.68–1.68, p = .774) |
| **Gender** | Female | 1443 (50.5%) | 352 (75.2%) | |
| | Male | 1415 (49.5%) | 116 (24.8%) | 0.52 (0.37–0.72, p < .001) |
| **Age** | Mean ± SD | 69.7 ± 8.8 | 73.3 ± 7.5 | 1.01 (0.99–1.03, p = .422) |
| **Address type** | City | 896 (31.4%) | 57 (12.2%) | |
| | Village | 1962 (68.6%) | 411 (87.8%) | 1.13 (0.74–1.73, p = .571) |
| **Work state** | Self-employed | 1678 (58.7%) | 388 (82.9%) | |
| | Employed | 1180 (41.3%) | 80 (17.1%) | 0.56 (0.40–0.79, p < .001) |
| **Martial status** | Unmarried | 226 (7.9%) | 93 (19.9%) | 0.47 (0.34–0.64, p < .001) |
| | Married | 2632 (92.1%) | 375 (80.1%) | |
| **Life satisfaction score** | Mean ± SD | 2.5 ± 0.7 | 1.9 ± 0.9 | 0.41 (0.36–0.48, p < .001) |
| **Social activity score** | Mean ± SD | 1.4 ± 1.0 | 1.1 ± 0.4 | 0.60 (0.47–0.76, p < .001) |
| **Smoke state** | Nonsmoker | 2023 (70.8%) | 377 (80.6%) | |
| | Smoker | 835 (29.2%) | 91 (19.4%) | 1.24 (0.89–1.72, p = .201) |
| **Drinking state** | Mean ± SD | 0.7 ± 0.9 | 0.4 ± 0.8 | 0.88 (0.76–1.02, p = .098) |
| **Sleep time** | Mean ± SD | 6.5 ± 1.8 | 5.2 ± 2.2 | 0.77 (0.73–0.81, p < .001) |
| **Previous work type** | Not Work | 349 (12.2%) | 76 (16.2%) | |
| | Non-agricultural Work | 1020 (35.7%) | 55 (11.8%) | 0.50 (0.32–0.79, p = .003) |
| | Agricultural Work | 1248 (43.7%) | 313 (66.9%) | 0.93 (0.67–1.29, p = .673) |
| | Both | 241 (8.4%) | 24 (5.1%) | 0.64 (0.38–1.09, p = .099) |

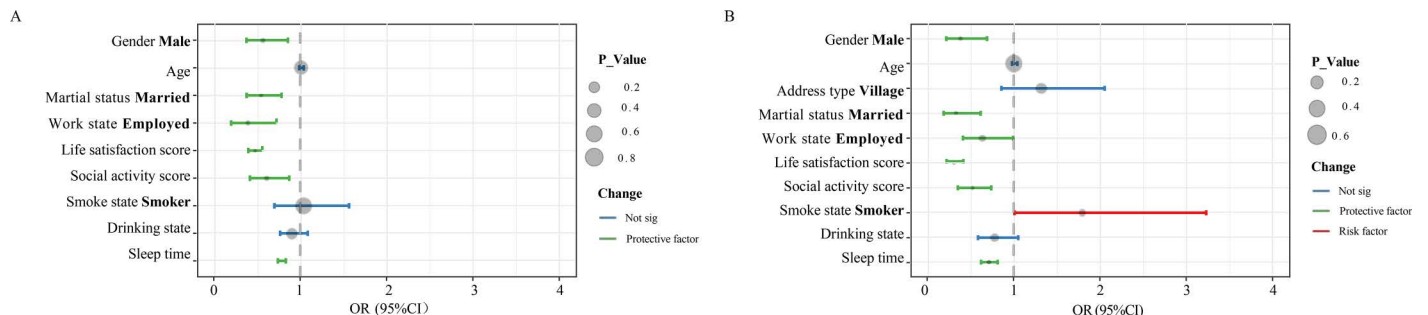

**Fig 4. Subgroup analysis by job type-risk factors for depressive states among agricultural and non-agricultural workers.**

to those working in agricultural jobs (OR = 1.80; 95% CI 1.01 to 3.20; p = 0.047). However, men (agricultural: OR = 0.56; 95% CI 0.37 to 0.85; p < 0.01 Non-agricultural (OR = 0.39; 95% CI 0.22 to 0.69; p < 0.001) and married (agricultural: OR = 0.54; 95% CI 0.37 to 0.69; p < 0.001) non-smokers were at greater risk than non-smokers. 95% CI 0.37 to 0.77; p < 0.001 Non-agricultural: OR = 0.33; 95% CI 0.18 to 0.60; p < 0.001), employed (agricultural: OR = 0.39; 95% CI 0.20 to 0.74; p < 0.01), and married (agricultural: OR = 0.39; 95% CI 0.20 to 0.74; p < 0.01). Non-agricultural: OR = 0.64; 95% CI 0.41 to 0.99; p < 0.05), life satisfaction (agricultural: OR = 0.47; 95% CI 0.39 to 0.56; p < 0.001 Non-agricultural: OR = 0.30; 95% CI 0.22 to 0.56; p < 0.001 95% CI 0.22 to 0.42; p < 0.001), participation in social activities (agricultural: OR = 0.61; 95% CI, 0.42–0.88; p < 0.01 Non-agricultural: OR = 0.52; 95% CI 0.36 to 0.76; p < 0.001), and sleep duration. 0.001) and

sleep duration (agricultural: OR = 0.78; 95% CI 0.74 to 0.83; p < 0.001 Non-agricultural: OR = 0.71; 95% CI 0.62 to 0.82; p < 0.001) were significant protective factors for both subgroups.

## Work state subgroups

Subjects were categorized into four subgroups, self-employed non-farmers (A), self-employed farmers (B), employed non-farmers (C), and employed farmers (D), based on the dual criteria of job states and job type, and the results of the logistic regression are shown in Fig 5.

As can be seen from the figure, in the group with non-agricultural jobs, self-employed subjects had only two significant protective factors for life satisfaction (OR = 0.24; 95%CI 0.14 to 0.42; p < 0.001) and sleep duration (OR = 0.75; 95%CI 0.60 to 0.93; p = 0.01) in comparison to employed subjects, while employed workers had only two significant protective factors in addition to life satisfaction (OR = 0.35; 95%CI 0.24 to 0.51; p < 0.001) and sleep duration (OR = 0.69; 95%CI 0.58 to 0.82; p < 0.001), there were two significant protective factors for employed workers in addition to male (OR = 0.28; 95%CI 0.12 to 0.63; p < 0.01), married (OR = 0.26; 95%CI 0.13 to 0.53; p < 0.001) and social activity participation (OR = 0.45; 95%CI 0.27 to 0.76; p < 0.01), which were three specific protective factors.

In contrast, in the agriculture group, there were no significant protective factors for employed subjects, while men (OR = 0.53; 95% CI 0.35 to 0.81; p < 0.01), married people (OR = 0.52; 95% CI 0.36 to 0.76; p < 0.001), life satisfaction (OR = 0.45; 95% CI 0.38 to 0.54; p < 0.001), social activity participation (OR = 0.64; 95% CI 0.43 to 0.95; p < 0.05), and sleep duration (OR = 0.78; 95% CI 0.73 to 0.82; p < 0.001) were significant protective factors for self-employed workers.

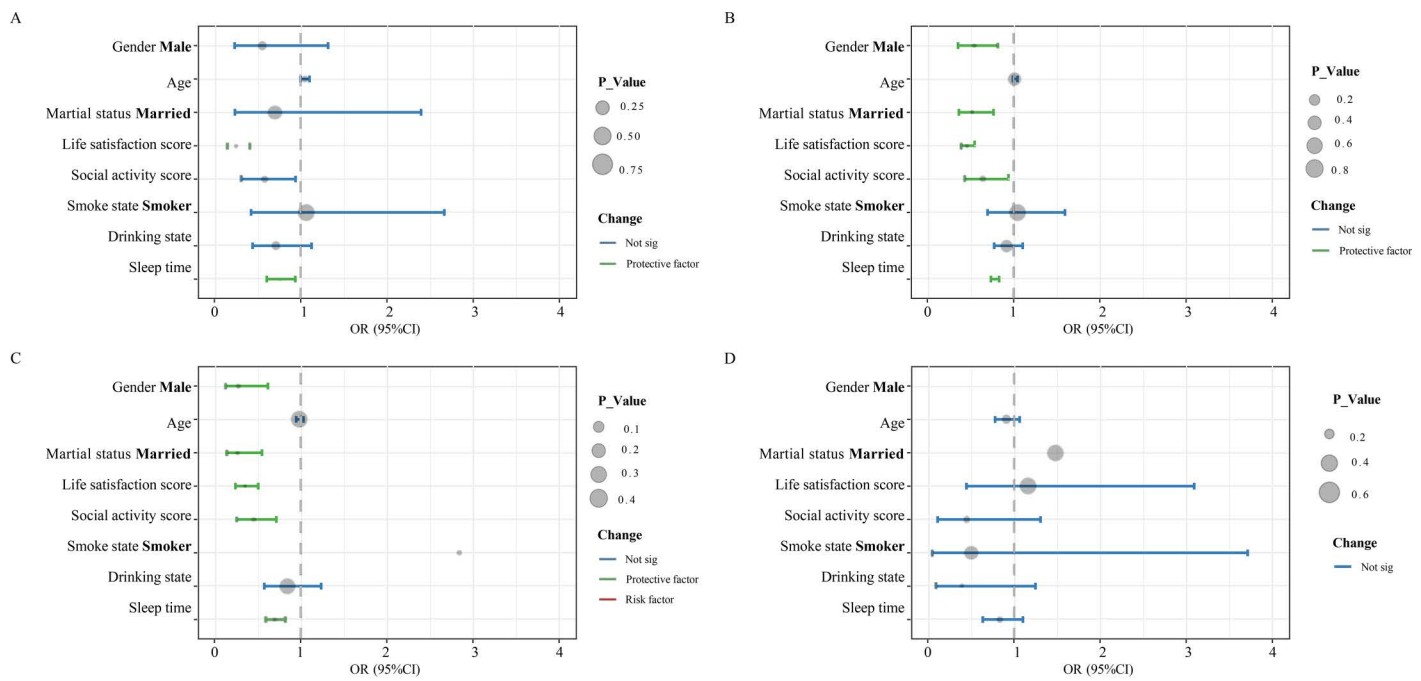

（A）Self-employed non-farmers subgroup
（B）Self-employed farmers subgroup
（C）Employed non-farmers subgroup
（D）Employed farmers subgroup

**Fig 5. Subgroup analysis by work state-risk factors for depressive states among self-employed and employed workers.**

## Mediation model analysis and test results

Baron and Kenny's stepwise regression was used to verify the mediating role of life satisfaction between job type (agricultural vs. non-agricultural) and depressive states, as shown in Table 4. The total effect of job type on depressive states (Path c) was significant (B = 3.622, p < 0.001). Life satisfaction scores showed a significant negative correlation with job type (Path a: B = −0.070, p < 0.001) and depressive states (Path b: B = −2.641, p < 0.001). The indirect effect mediated by life satisfaction (Path a*b) was 0.186 (SE = 0.070, 95% CI 0.0492 to 0.320, p < 0.01), indicating a significant indirect effect. After controlling for the mediator variable life satisfaction, the coefficient of job type on depressive states (Path c': B = 3.437, p < 0.001) remained significant, indicating that the direct effect was also significant and that the effect of job type on depressive states were partially mediated by life satisfaction.The final mediation model is shown in Fig 6.

As shown in Table 5, the estimate of ACME (Average Causal Mediation Effects) was 0.1858, indicating that life satisfaction as a mediating variable produced a significant positive indirect effect between job type and depressive state, suggesting that agricultural workers increased their depressive state by 0.1858 standard deviations through life satisfaction. The mediating effect explained 5.13% of the total effect (B = 3.6224, p < 0.001), suggesting that the level of life satisfaction scores played a role in explaining the role of job type on depressive states.

The sensitivity analysis results are shown in Figure Fig 7. The X-axis is the sensitivity parameter Rho and the Y-axis is the value of the mediating effect. When the point estimate of the average causal mediating effect ACME is 0, i.e., the absolute value of rho when the mediating effect disappears is 0.3, which indicates that the confounding effect is stronger, i.e., the results of this mediating effect model are more reliable.

## Discussion

This cross-sectional study looks at the effect of job type, whether or not one works in agriculture on depressive states among middle-aged and older adults over the age of 45. Based on a series of regression and mediation analyses from the

**Table 4. Mediation analysis.**

| Variable | Path c | | Path a | | Path c' and Path b | | Path a*b | | | |
|---|---|---|---|---|---|---|---|---|---|---|
| | B | SE | B | SE | B | SE | B | SE | LLCI | ULCI |
| | 3.622*** | 0.213 | −0.070*** | 0.026 | 3.437*** | 0.201 | 0.186** | 0.070 | 0.0492 | 0.320 |

Note:*p < 0.05;**p < 0.01; ***p < 0.001

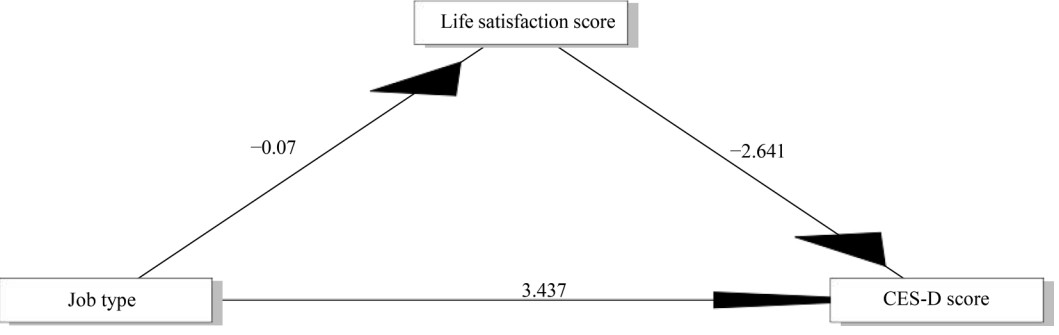

**Fig 6. Mediation model.**

**Table 5. Causal mediation analysis.**

|  | Estimate | 95% CI Lower | 95% CI Upper | p-value |
|---|---|---|---|---|
| ACME | 0.1858 | 0.0492 | 0.32 | 0.002** |
| ADE | 3.4366 | 3.0224 | 3.84 | < 0.001*** |
| Total Effect | 3.6224 | 3.2044 | 4.03 | < 0.001*** |
| Prop.Mediated | 0.0513 | 0.0141 | 0.09 | 0.002** |

Note:*p<0.05; **p<0.001

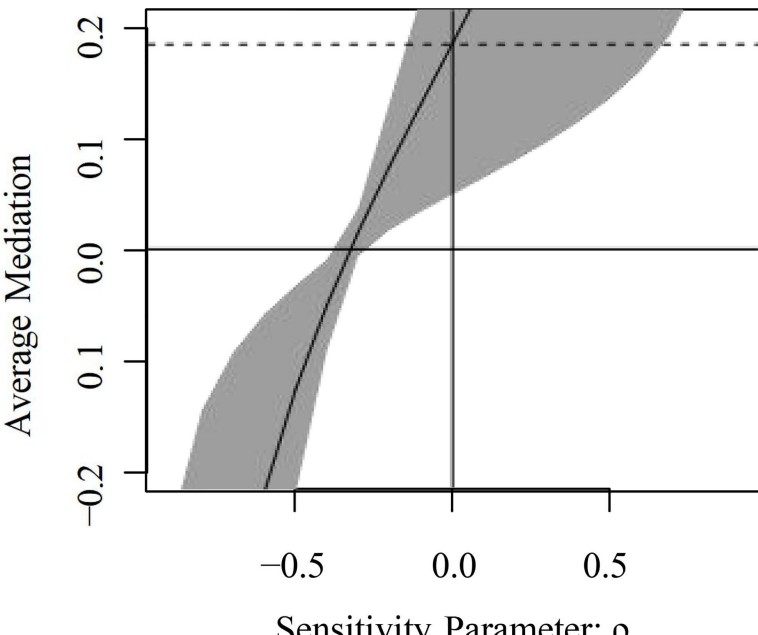

**Fig 7. Sensitivity analysis.**

CHARLS national survey, the findings suggest that agricultural workers have significantly higher depression scores than non-agricultural workers, while life satisfaction partially mediates the association between job type and depression.

Our findings indicate that subjects working in agriculture had higher mean CES-D scores than those working in non-agricultural jobs, suggesting that depressive states are more severe in agricultural jobs, which is not consistent with our hypothesis based on horticultural therapy. Through further Logistic multifactorial regression analysis, we could observe that the type of jobs did not play a role in the occurrence of depressive states under the consideration of the inclusion of other covariates, suggesting that the facilitating effect of agricultural work on depressive states was in the presence of other exposure factors. Previous studies have indicated that exposure factors such as pesticide exposure [39–41], social activities [42], and social support [43] are associated with depressive states among agricultural workers, whereas in the present study, the researcher's baseline showed that life satisfaction scores and social activity participation scores of agricultural workers were lower than those of non-agricultural workers, and further subgroup regression analyses of the type of job with or without farming it can be observed that the increase in life satisfaction and social activity participation of agricultural workers reduces the risk of the occurrence of depressive states. Thus, we can propose the idea that lower life satisfaction and social activity participation are risk factors for the development of depression among agricultural

workers. The reasons for low life satisfaction among agricultural workers are complex, primarily influenced by the following factors:(1) Economic Pressure [41,44]: Agricultural workers' incomes are generally less stable compared to those in non-agricultural occupations, as they are highly susceptible to weather changes, market fluctuations, and other external factors. The seasonal and uncertain nature of agricultural income can impose substantial financial stress, leading to lower life satisfaction.(2) Occupational Health Risks [45]: Agricultural workers face high occupational health risks, including long working hours, exposure to adverse weather conditions, and pesticide exposure mentioned earlier. These health risks negatively impact both physical and mental well-being, reducing overall life satisfaction.(3) Social Isolation and Lack of Social Support [44]: Agricultural workers are often geographically isolated due to the nature of their work, resulting in limited social interactions and relationships. This isolation contributes to lower levels of social participation. The lack of social support and limited opportunities for interaction may lead to feelings of loneliness and depressive symptoms, further impacting life satisfaction.(4) Mental Health Issues and Cultural Stigma [45]: In rural areas, mental health issues are still subject to cultural stigma, which makes it difficult for agricultural workers experiencing mental health challenges to seek help. The lack of mental health support resources can exacerbate these issues, leading to further declines in life satisfaction.

The baseline shows that compared to non-agricultural workers, agricultural workers live in rural areas. Urbanization has been a trend in the world, and China has followed the world trend and entered the process of rapid urbanization [46–48], which has led to the center of gravity of socio-economic activities being located in the city, resulting in a dramatic change in living environments and living factors. Therefore, the difference between urban and rural life affects mental health to a certain extent, and according to existing studies [49,50] we can point out that the main reason for this effect is that the difference between urban and rural living space and socio-economics makes urban areas have more diversified social activities and more abundant social resources compared to rural areas, which leads to a significant increase in the life satisfaction of urban residents. The social resources available in cities include education [49], health care services [51], pension benefits [49], social infrastructure, and so on. A high level of education can make middle-aged and elderly people have good adaptability in the face of adversity, and has been shown that the education of middle-aged and elderly people will affect the education of their own offspring, which in turn affects the intergenerational support for the elderly, and thus has an impact on the symptoms of depression among the elderly [52];The lack of medical resources in rural areas prevents middle-aged and elderly people from seeking timely medical treatment when they face a series of health problems, which can lead to functional disabilities, and it has been pointed out in an empirical study that functional disabilities of middle-aged and elderly people reduce life satisfaction and aggravate their depressive symptoms [16]; health insurance and pension insurance coverage is higher in urban areas, which can alleviate the economic pressure to a certain extent. This suggests that living in cities helps to increase life satisfaction [53]. There is also evidence from Asia that participation in social activities to increase social engagement has a positive impact on the mental health of middle-aged and older adults [42,51].Current studies indicate a significant negative correlation between digital village construction and depressive symptoms among rural middle-aged and older adults, with digital village construction alleviating depression through income, consumption, and cognitive welfare effects [14]. Based on these findings, the following measures can effectively leverage the three welfare effects of digital village construction to reduce depressive risks among rural agricultural workers and promote healthy aging: 1) enhancing digital infrastructure (e.g., improving rural internet coverage and e-commerce platforms to boost income) [54]; 2) organizing digital literacy training to strengthen cognitive abilities [55,56]; 3) optimizing logistics and online services to facilitate consumption [57].

Our findings suggest that males are a protective factor for the occurrence of depressive states in both the farming and non-farming populations, indicating that there is a gender difference in the occurrence of depressive states and that females are at a higher risk for their occurrence, which is consistent with the findings of previous studies [58–60]. There are many factors that contribute to the high prevalence of depressive states in the female population, including a combination of biological, psychological, and sociocultural factors. Biologically, women are more sensitive to hormonal fluctuations

associated with the reproductive cycle, such as hormonal fluctuations (menstrual cycle, postpartum, and menopause) that can affect mood thus leading to the occurrence of premenstrual irritability disorder and postpartum depressive states, among others [59–61]; psychology points out that women are more prone to recurring thoughts of negativity [59], which may be associated with an increased risk of depressive states; and in the process of growing up, the Gender roles and societal expectations place additional stress on women, with sexual violence [62], sexual abuse [60], interpersonal violence, stress exposure and stress susceptibility [33] at the micro level, and gender inequality [63,64] at the macro level, all of which increase the risk for the development of depressive states in women. Marital states can also have a significant impact on an individual's emotional health, and there have been precedent results demonstrating that marriage is a protective factor for depressive states [65–67], a finding that is consistent with our findings. There is a positive correlation between marriage and mental health, the main reason is that marriage provides emotional fulfillment and social support, the intimate relationship between marriage partners can provide each other with emotional support and encouragement, help each other to cope with setbacks and difficulties in life, reduce loneliness and social isolation, and enhance mental health. In addition to this marriage implies the sharing of economic resources such as housing, income and property, which can reduce financial stress and anxiety. However, it is worth noting that marital quality plays an important role in moderating the effects of marriage on mental health [65].

In order to better provide targeted recommendations for the adjustment of workers' depressive state, we further conducted subgroup regression analysis based on whether or not they were self-employed, and the results showed that in the agricultural population, self-employed people had more protective factors than employed people, while in the non-agricultural population, employed people had more protective factors than self-employed people. According to the conclusion of the previous study [58], self-employed workers have a greater risk of depression. We analyze that the causes of depression among self-employed workers may lie in the following points: 1) Unstable source of economic income: self-employed workers, whether they work in agriculture or non-agriculture, need to be responsible for their own work profit and loss, and the economic pressure is high, while in comparison with the employed workers' source of economic income is more stable, with better welfare benefits such as health insurance and health care, and the benefits are better. better welfare benefits, such as higher coverage of health insurance and pension insurance [68]; 2) Blurred work-life boundaries: the uncertainty of working hours often takes up time for private life, resulting in insufficient time for rest and relaxation; 3) High work intensity: in order to maintain continuity and development of their work business, self-employed workers need to spend more time and energy than employed workers; 4) Self-employment is mostly in agriculture, construction, mining, etc., and poorer workplace exposure exposes workers to higher physical and mental health risks [58,69]; 5) And self-employed workers usually lack social support from organizations, which also increases their risk of depression [58]. For the agricultural self-employed and non-agricultural employed people, there are several suggestions we proposed: 1) Improve life satisfaction by improving the quality of life [70,71]; 2) Participate in more social activities to enhance the interaction with friends and social support [42,43,51]; 3) Extend the sleep time and develop good sleep habits to get sufficient physical and mental rest and relaxation [72–74]. Moreover, for the government to complement public health policies for agricultural workers, we proposed the following suggestions: 1) Construct more rural fitness and recreational facilities; 2) Conduct more mental health screening activities in rural areas; 3) Implement policies to improve the incomes of agricultural workers in rural areas.

According to the results of BK stepwise regression, it can be observed that there is a partial mediating effect of life satisfaction between the level of depressive state and the type of jobs, i.e., the negative correlation between life satisfaction and the type of jobs indicates that the life satisfaction of agricultural workers is lower than that of non-agricultural workers, and the negative correlation between life satisfaction and depressive state indicates that the risk of depressive state can be reduced by improving life satisfaction, which indicates that agricultural workers have low life satisfaction due to low depressive state. The negative correlation between life satisfaction and depression indicates that the risk of depression can be reduced by increasing life satisfaction, suggesting that agricultural workers are depressed due to low

life satisfaction. In terms of the mechanism of depression, studies have shown that there is a close association between depression and life satisfaction, and that the occurrence of depressive symptoms is often accompanied by a decrease in life satisfaction [75], and people with low life satisfaction are more likely to experience depression [76]; the results of a priori RCT have shown that depression symptoms can be significantly reduced by increasing life satisfaction [50], and the results of another RCT found that an active psychological intervention, if conducted for a year, was effective in reducing depressive symptoms and increasing life satisfaction [75]. All of these findings further support the results of our mediation analysis and are consistent with the findings of the regression analysis.

This study has several limitations. First, it is based on cross-sectional data from the 2015 China Health and Retirement Longitudinal Study (CHARLS), which limits the ability to establish a clear temporal sequence between agricultural employment and depressive symptoms. Consequently, reverse causality cannot be ruled out; for instance, individuals with depression may be more inclined to engage in agricultural work. Longitudinal designs are essential for understanding the temporal relationship between job type and mental health outcomes. Existing longitudinal investigations have illuminated that depressive symptoms are not static; rather, they can fluctuate over time, which cross-sectional studies may not adequately represent. Longitudinal research highlights the dynamic relationship between job type and mental health, with studies showing that unstable employment can lead to increased depressive symptoms over time [77] and work-related stressors can exacerbate mental health issues [78]. Future research should employ longitudinal data to more accurately elucidate causal relationships between variables. Second, a substantial number of samples (n = 11,270) were excluded due to missing covariate data, which may introduce selection bias. Future studies should consider employing multiple imputation techniques to handle missing data and conduct sensitivity analyses to assess the robustness of the findings. Additionally, the classification of occupational types was simplistic, dividing them merely into "agricultural" and "non-agricultural," failing to capture the heterogeneity within agricultural work, such as subsistence versus commercial farming and land ownership differences. Future research should refine occupational classifications to more comprehensively understand the impact of occupational types on depressive symptoms. The study also employed logistic regression to analyze depressive symptoms, a common outcome variable, which may overestimate odds ratios. Future studies might consider alternative regression models, such as Poisson regression, to obtain more accurate estimates. Although this study found that life satisfaction had a mediating effect between agricultural work and depressive symptoms, the effect was weak, accounting for only 5.13% of the total effect. This may be due to the omission of other potential mediating variables, such as economic stability and pesticide exposure. Future research should explore additional mediators to more fully understand the mechanisms by which agricultural work influences depressive symptoms. Finally, the 2015 data used in this study may not capture recent socioeconomic changes in rural China, such as digital development and policy shifts. Additionally, mental health research involves sensitive information, necessitating careful ethical considerations to ensure participant privacy and informed consent during data collection and analysis. Future studies should focus on the temporal relevance of data and enhance ethical review and participant protection measures.

## Conclusion

Our study investigated the relationship between the type of jobs, whether farming or not, and depressive states, emphasizing through subgroup analysis that self-employed job type, agricultural job and sex as female are risk factors for developing depressive states and developing depression. In addition, life satisfaction mediated the relationship between the type of job and the level of depressive state, a result that highlights the complex association between the type of job and depressive state.

Therefore, based on the results of our logistic regression and mediation analyses, public health recommendations that can intervene in the occurrence of depressive states or alleviate existing depressive states by improving life satisfaction can be made for agricultural workers in order to help agricultural workers have a healthier psychological state. Future

research could focus on the mechanisms by which differences in specific parameters of life satisfaction under different job type classifications affect depressive states and whether other mediating variables are also present.

## Author contributions

**Conceptualization:** Feiying He, Xiaoying Li, Hualin Liu, Yanwu Chen, Yuan Yao Chen.

**Data curation:** Xiaoying Li, Xiangchun Xu, Shulin Bao, Yanwu Chen.

**Formal analysis:** Xiaoying Li, Xiangchun Xu, Shulin Bao.

**Software:** Hualin Liu.

**Supervision:** Feiying He, Xiangchun Xu, Yuan Yao Chen.

**Validation:** Feiying He, Xiaoying Li.

**Writing – original draft:** Xiaoying Li, Xiangchun Xu.

**Writing – review & editing:** Feiying He, Hualin Liu, Yuan Yao Chen.

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
