## [Decision Letter · Decision Letter 0]

PONE-D-24-33298Can working in agriculture have a favorable effect on depressive symptoms?Life satisfaction as a mediatorPLOS ONE

Dear Dr. Chen,

Thank you for submitting your manuscript to PLOS ONE. After careful consideration, we feel that it has merit but does not fully meet PLOS ONE’s publication criteria as it currently stands. Therefore, we invite you to submit a revised version of the manuscript that addresses the points raised during the review process.

**I hope this message finds you well. Thank you for submitting your manuscript, "Can Working in Agriculture Have a Favorable Effect on Depressive Symptoms? Life Satisfaction as a Mediator," to PLOS ONE. I have carefully reviewed your work along with the comments provided by our esteemed reviewers. While all three reviewers have recommended minor revisions, upon careful consideration of their feedback and a thorough evaluation of your manuscript, I believe that a more comprehensive set of changes—what we typically categorize as a major revision—would significantly enhance the quality and clarity of your paper.**

We look forward to receiving your revised manuscript.

Kind regards,

Wenbin Du

Academic Editor

PLOS ONE

Funding

This research is supported by Key Laboratory of Philosophy and Social Sciences of Colleges and Universities in Guangdong Province (2023WSYS005, 2023WSYS007), the 13th Five-Year Plan of Guangdong Province for Philosophy and Social Sciences (GD20XGL42).

3. In the online submission form, you indicated that your data is available only on request from a third party. Please note that your Data Availability Statement is currently missing the name of the third party contact or institution. Please update your statement with the missing information. 

Reviewers' comments:

Reviewer's Responses to Questions

**Comments to the Author**

1. Is the manuscript technically sound, and do the data support the conclusions?

Reviewer #1: Yes

Reviewer #2: Yes

Reviewer #3: Yes

2. Has the statistical analysis been performed appropriately and rigorously? 

Reviewer #1: Yes

Reviewer #2: Yes

Reviewer #3: Yes

3. Have the authors made all data underlying the findings in their manuscript fully available?

Reviewer #1: Yes

Reviewer #2: Yes

Reviewer #3: Yes

4. Is the manuscript presented in an intelligible fashion and written in standard English?

Reviewer #1: Yes

Reviewer #2: Yes

Reviewer #3: Yes

5. Review Comments to the Author

**Reviewer #1:**  1. Add 'China' to the keywords to improve the searches of the article when published on online databases.

2. In the introduction, it is important to expose the research gaps by reviewing related studies in the Chinese context especially. While the global stretch of this have been done, only a study referencing China has been cited though there are a lot of studies in China that have investigated the correlation between job(s) and depression with life satisfaction playing a mediating role.

3. It's important to reference some of the studies that have validated the CES-D scale with much success, pointing to it as reliable and as such efficacious for such a study.

4. In the selection of the covariates, do well to reference the previous studies that were consulted and justify the covariates that were settled on for the study.

5. In the discussion, the proposition put forward which is based on the significant results attained after running rigorous analysis, which are lower life satisfaction and social activity participation as major risk factors for the development of depression among agricultural workers need to be discussed from the perspective of other empirical studies. Which existing studies on work-related depression have recorded similar results? What might probably be the underlying causative factors for recording such results? Discussion of these questions must be expanded in the discussion section. This was beautifully done in the discussion on the high risk of depression recorded among self-employed agricultural workers.

6. The limitations must be moved from the discussion to the concluding section. Also, after summarizing the study's purpose, it is important to reiterate the key results and draw valid conclusions from them based on the underlying research questions and hypotheses set. This has not been done.

**Reviewer #2: ** The study “Can working in agriculture have a favorable effect on depressive symptoms? Life satisfaction as a mediator” investigates the impact of agricultural work on depressive symptoms and whether life satisfaction mediates this relationship. The study concluded that self-employed workers in agriculture have a higher risk of developing depressive states and present a high risk of developing depression. This is a relevant finding for research related to depressive disorders. However, there are areas that require clarification and significant improvements, particularly regarding the background, research design development, data analysis, and alignment of the research questions with the results. Addressing these points would strengthen the overall impact and credibility of the work. Notably, the manuscript is generally well-written, and the authors should be able to address my points through a revision. My comments and suggestions for improvement are described below.

Introduction

2nd paragraph:

In the excerpt: "...but there is still no research focusing on the association between job type and depression…"

There is an inconsistency because the two references below associate depression and job type:

1-) Kim S, Kwon M, Seo K. Factors Influencing the Health-Related Quality of Life of Workers According to the Type of Work. Healthcare (Basel). 2022 Oct 18;10(10):2066. doi: 10.3390/healthcare10102066;

2-) Kang W, Park WJ, Jang KH, Lim HM, Ann JS, Cho SH, Moon JD. Comparison of anxiety and depression status between office and manufacturing job employees in a large manufacturing company: a cross sectional study. Ann Occup Environ Med. 2016 Sep 15;28:47. doi: 10.1186/s40557-016-0134-z.

2nd and 3rd paragraphs: In the excerpts below:

“This study will then look at different types of work - agricultural and non-agricultural - and explore the association with depression, and further analyze subgroups of the population according to whether they are self-employed or employed".

“Therefore, in this study, it is necessary to investigate whether the type of jobs has a correlation with life satisfaction and to include life satisfaction in the model as a mediating variable to investigate whether the type of jobs acts on the depressive state through life satisfaction”.

I suggest focusing the research objectives in a single paragraph, preferably in the last paragraph of the Introduction.

3nd paragraph: In the excerpt: " ...It has been proven that there is an association between life satisfaction and depression in middle-aged and elderly people."

It is not clear what type of association exists between the two mentioned variables.

Method

Research design

I suggest specifying the method used for the sample calculation.

Study population

I suggest reporting the exclusion criteria applied to the study.

Measurement Tools

Life satisfaction

In the excerpt: “Life satisfaction was assessed at baseline by asking participants: How satisfied are you with your life in general?"

Did the participants receive any information about what life satisfaction is and/or some more objective criteria to assist in their self-analysis?

I believe that a subjective question may lead participants to potential interpretation biases, making it necessary to establish criteria for them to respond more accurately.

Methods of statistical analysis

It is not clear what covariates were included in the correlation analysis.

Result/Discussion

I suggest reporting the influence of working hours and its relationship with depression and life satisfaction of agricultural workers.

Discussion

4nd paragraph:

In the excerpt: “There are many factors that contribute to the high prevalence of depressive states in the female population, including a combination of biological, psychological, and sociocultural factors. Biologically, women are more sensitive to hormonal fluctuations associated with the reproductive cycle, such as hormonal fluctuations (menstrual cycle, postpartum, and menopause) that can affect mood thus leading to the occurrence of premenstrual irritability disorder and postpartum depressive states, among others[32 –34]; psychology points out that women are more prone to recurring thoughts of negativity[32], which may be associated with an increased risk of depressive states; and in the process of growing up, the Gender roles and societal expectations place additional stress on women, with sexual violence [35], sexual abuse [33], interpersonal violence, stress exposure and stress susceptibility [33] at the micro level, and gender inequality[36, 37] at the macro level, all of which increase the risk for the development of depressive states in women.”

What is the relationship between the mentioned factors and depression associated with agricultural work?

Conclusion

I suggest reporting the relevance of the research for public health policies in China. In light of the study's results, what can be done to improve the quality of life for agricultural workers?

**Reviewer #3:**  1. The study addresses an important and understudied area: the relationship between agricultural work and depressive symptoms. However, the use of a single-item measure for life satisfaction, while common, may not capture the full complexity of the construct. The study doesn't fully explore potential mechanisms underlying the relationship between agricultural work and depressive symptoms.

2. the article presents a well-structured study with a solid methodology and employing appropriate statistical techniques for the research questions. The use of propensity score matching to balance the agricultural and non-agricultural groups is a strength. The results are computed using appropriate statistical techniques, including logistic regression analysis to identify protective factors associated with life satisfaction and depressive symptoms. The use of odds ratios (OR) with 95% confidence intervals (CI) provides a clear understanding of the strength and significance of the associations. Additionally, mediation analysis is conducted to explore the role of life satisfaction in the relationship between job type and depressive symptoms, which is a robust approach to understanding causal pathways. The sensitivity analysis adds another layer of reliability to the findings, indicating that the mediating effects are not overly influenced by confounding variables. Overall, the statistical methods employed are appropriate for the research questions posed. However, there are some potential drawbacks:

• The dichotomization of the CES-D scores for logistic regression may lead to loss of information.

• The study doesn't address potential selection bias - people with certain characteristics may be more likely to engage in agricultural work.

3. The finding that agricultural workers have higher depressive symptom scores is surprising and doesn't align with some existing literature on the potential mental health benefits of nature-based activities. However, this could be due to other factors associated with agricultural work in China, such as economic stress or physical strain. The mediating role of life satisfaction is consistent with previous research on well-being and mental health. The subgroup analyses provide interesting insights into how the relationships differ across various populations, which adds nuance to the overall findings.

4. There are numerous grammatical and typographical errors throughout the text, which should be carefully edited.

6. PLOS authors have the option to publish the peer review history of their article (what does this mean? ). If published, this will include your full peer review and any attached files.

**Do you want your identity to be public for this peer review?** For information about this choice, including consent withdrawal, please see our Privacy Policy .

Reviewer #1: **Yes: ** Dickson Adom

Reviewer #2: **Yes: ** Débora Regina de Aguiar

Reviewer #3: No

---

## [Author Response · Author response to Decision Letter 1]

28 Nov 2024

Dear Reviewer 1,

Thank you very much for your comments. I have read your comments carefully and made a modification to the original manuscript. The revised part was marked in yellow based on your comments. The following are my responses to your questions.

Reviewer #1: 1. Add 'China' to the keywords to improve the searches of the article when published on online databases.

Our response: Thank you for your kind comment and advise, we have revised and added the key word “China” into our key words.(It can be reviewed in the Key Words part in our revised manuscript in yellow):

“Key Words: Agriculture work; Depressive symptoms; Life satisfaction; CHARLS; China”

2. In the introduction, it is important to expose the research gaps by reviewing related studies in the Chinese context especially. While the global stretch of this have been done, only a study referencing China has been cited though there are a lot of studies in China that have investigated the correlation between job(s) and depression with life satisfaction playing a mediating role.

Our response: Thank you for your valuable feedback. Our research was conducted within the context of China, and it was an oversight on our part not to cite more studies specific to China. We have now included several references that examine the relationship between work and depressive symptoms within the Chinese context, highlighting the mediating role of life satisfaction,as follows(At Line 62-64 in revised manuscript):

“Research has shown that work is closely related to the development of depression[4], which can be subdivided into work pressure[5], work hours[6, 7] , work environment[8, 9],work support[10],subjective social status[11]and so on,...”

3. It's important to reference some of the studies that have validated the CES-D scale with much success, pointing to it as reliable and as such efficacious for such a study.

Our response: Thank you for your valuable comment, it is our flaw to make you confuse about our validity of the CES-D scale. We have cited previous researches about the scale to evident its validity and reliability, which is as follows(At Line 126 in revised manuscript):

“... ,which is now a widely used and proven valid and reliable common indicator of depression[19, 20].”

4. In the selection of the covariates, do well to reference the previous studies that were consulted and justify the covariates that were settled on for the study.

Our response: Thank you for your valuable comment.We have cited previous studies to demonstrate the relationships between each covariate and the dependent variable (depressive symptoms), thereby supporting the rationale for our choice of covariates.

5. In the discussion, the proposition put forward which is based on the significant results attained after running rigorous analysis, which are lower life satisfaction and social activity participation as major risk factors for the development of depression among agricultural workers need to be discussed from the perspective of other empirical studies. Which existing studies on work-related depression have recorded similar results? What might probably be the underlying causative factors for recording such results? Discussion of these questions must be expanded in the discussion section. This was beautifully done in the discussion on the high risk of depression recorded among self-employed agricultural workers.

Our response: Thank you for your valuable comment.We have expanded our discussion and thoroughly restructured the logical flow in Discussion section,as follows(At Line 329-344 in revised manuscript):

“The reasons for low life satisfaction among agricultural workers are complex, primarily influenced by the following factors:(1) Economic Pressure[41]: Agricultural workers’ incomes are generally less stable compared to those in non-agricultural occupations, as they are highly susceptible to weather changes, market fluctuations, and other external factors. The seasonal and uncertain nature of agricultural income can impose substantial financial stress, leading to lower life satisfaction.(2) Occupational Health Risks[42]: Agricultural workers face high occupational health risks, including long working hours, exposure to adverse weather conditions, and pesticide exposure mentioned earlier. These health risks negatively impact both physical and mental well-being, reducing overall life satisfaction.(3) Social Isolation and Lack of Social Support[41]: Agricultural workers are often geographically isolated due to the nature of their work, resulting in limited social interactions and relationships. This isolation contributes to lower levels of social participation. The lack of social support and limited opportunities for interaction may lead to feelings of loneliness and depressive symptoms, further impacting life satisfaction.(4) Mental Health Issues and Cultural Stigma[42]: In rural areas, mental health issues are still subject to cultural stigma, which makes it difficult for agricultural workers experiencing mental health challenges to seek help. The lack of mental health support resources can exacerbate these issues, leading to further declines in life satisfaction.”

6. The limitations must be moved from the discussion to the concluding section. Also, after summarizing the study's purpose, it is important to reiterate the key results and draw valid conclusions from them based on the underlying research questions and hypotheses set. This has not been done.

Our response Thank you for your valuable advise and comment, we have adjusted the narration order, revised the language and moved it to the conclusion section. For helping you better review our revised part, the revised part is at conclusion section Line 413-420 of revised manuscript.

Best Regards,

Yuan Yao Chen

yuanyaochen0401@163.com

Dear Reviewer 2,

Thank you very much for your comments and appreciation. We have read your comments carefully and made a modification to the original manuscript. The revised part was marked in yellow based on your comments. The following are my responses to your questions.

Reviewer #2: The study “Can working in agriculture have a favorable effect on depressive symptoms? Life satisfaction as a mediator” investigates the impact of agricultural work on depressive symptoms and whether life satisfaction mediates this relationship. The study concluded that self-employed workers in agriculture have a higher risk of developing depressive states and present a high risk of developing depression. This is a relevant finding for research related to depressive disorders. However, there are areas that require clarification and significant improvements, particularly regarding the background, research design development, data analysis, and alignment of the research questions with the results. Addressing these points would strengthen the overall impact and credibility of the work. Notably, the manuscript is generally well-written, and the authors should be able to address my points through a revision. My comments and suggestions for improvement are described below.

Our response Thank you for your appreciation, valuable comments and suggestions on our research, we have carefully read and responded to your comments as following responses. We have fixed and revised most of the problems you mentioned and still tried to improve our future study based your comments.

Introduction

1.2nd paragraph:

In the excerpt: "...but there is still no research focusing on the association between job type and depression…"

There is an inconsistency because the two references below associate depression and job type:

Our response Inappropriate wording in our article was an oversight on our part. Our paper examines the relationship between two types of jobs—agricultural and non-agricultural work which we have made an explanation at Method—and depressive symptoms. However, the definitions of “job type” in the two references below differ from ours. We are very grateful for your feedback on this matter and have made the following revisions accordingly (At Line 65 in revised manuscript):

“but there is still no research focusing on the association between job type—agricultural and non-agricultural— and depression.”

2.“This study will then look at different types of work - agricultural and non-agricultural - and explore the association with depression, and further analyze subgroups of the population according to whether they are self-employed or employed".

“Therefore, in this study, it is necessary to investigate whether the type of jobs has a correlation with life satisfaction and to include life satisfaction in the model as a mediating variable to investigate whether the type of jobs acts on the depressive state through life satisfaction”.I suggest focusing the research objectives in a single paragraph, preferably in the last paragraph of the Introduction.

3nd paragraph: In the excerpt: " ...It has been proven that there is an association between life satisfaction and depression in middle-aged and elderly people."

It is not clear what type of association exists between the two mentioned variables.

Our response Thank you for your valuable comment.

(1)We have made revisions and consolidated the study objectives in the final paragraph of the introduction, as follows(At Line 79-83 in revised manuscript):

“This study will first examine the association between different types of work and depressive symptoms. Further, it will analyze population subgroups based on work state—self-employed versus employed—in order to provide more targeted public health recommendations. Additionally, life satisfaction will be incorporated into the model as a mediating variable to explore whether job types influence depressive symptoms through the pathway of life satisfaction.”

(2)The two referenced studies[14,15] reveal that previous research has established a correlation between depressive symptoms and life satisfaction. Reference [14] indicates that life satisfaction can serve as a potential indicator of depression, although it may be influenced by gender differences. Reference [15] highlights a reciprocal and enduring relationship between life satisfaction and depression.

Method

2.Research design

I suggest specifying the method used for the sample calculation.

Our response Thank you for your valuable comment, it is our flaw that the method used for the sample calculation is not explained enough. Firstly, the data we use for analysis in this article comes from the public database CHARLS,which are covered in detail in the Method_ Research design section.For the convenience of your review, we have modified the flow chart (Fig.1) to make you more clear about the method used in our sample calculation. Additionally,We have added our methodology for data analysis as follows(At Line 173 in revised manuscript):

“The above statistical analysis is based on R-4.3.1.”

3.Study population

I suggest reporting the exclusion criteria applied to the study.

Our response Thank you for your valuable comment.In our study, the criteria for inclusion of subjects at baseline were: subjects who contained a record of jobs type , assessment of depressive states, assessment of life satisfaction, age, gender, type of residence, work states, marital states, socially engaged activity score, smoking states, drinking states, sleep duration, prior work states in CHARLS 2015. Meanwhile, we have excluded subjects that have missing values in the variables given upon.We have explained this in the Methods_Study population section of the manuscript.For the convenience of your review, we have modified the flow chart (Fig.1) to make you more clear about our exclusion criteria.

4.Measurement Tools

Life satisfaction

In the excerpt: “Life satisfaction was assessed at baseline by asking participants: How satisfied are you with your life in general?"

Did the participants receive any information about what life satisfaction is and/or some more objective criteria to assist in their self-analysis?

I believe that a subjective question may lead participants to potential interpretation biases, making it necessary to establish criteria for them to respond more accurately.

Our response Thank you for your valuable comment.The data we use for analysis in this article comes from the public database CHARLS.There is a question "How satisfied are you with your life in general?" included in the questionnaire of CHARLS,which have been proved that strongly correlates with the validated 5-item Satisfaction with Life Scale (SWLS).We have cited the previous research[21] to explain this correlation.

5.Methods of statistical analysis

It is not clear what covariates were included in the correlation analysis.

Our response Thank you for your valuable comment.The covariates included in our study cover demographic characteristics such as age, gender, type of residential address and marital states, as well as factors that have been validated by research to be highly correlated with depressive states, such as social activity participation, smoking states, drinking states.We have illustrated in Method_Selection of covariates and shown in Fig.3.

Result/Discussion

6.I suggest reporting the influence of working hours and its relationship with depression and life satisfaction of agricultural workers.

Our response Thank you for your valuable comment.Certainly, as you mentioned, this is a very worthwhile area of study. However, because we used a public database, the questionnaire design in this database did not include questions regarding ‘working hours.’ Consequently, we lack a data source on working hours, which limits our ability in this study to explore the effects of working hours and their relationship with depressive symptoms and life satisfaction among agricultural workers. We are eager and will strive to incorporate this direction into future research as we undertake independent studies.

Discussion

7.4nd paragraph:

In the excerpt: “There are many factors that contribute to the high prevalence of depressive states in the female population, including a combination of biological, psychological, and sociocultural factors. Biologically, women are more sensitive to hormonal fluctuations associated with the reproductive cycle, such as hormonal fluctuations (menstrual cycle, postpartum, and menopause) that can affect mood thus leading to the occurrence of premenstrual irritability disorder and postpartum depressive states, among others[32 –34]; psychology points out that women are more prone to recurring thoughts of negativity[32], which may be associated with an increased risk of depressive states; and in the process of growing up, the Gender roles and societal expectations place additional stress on women, with sexual violence [35], sexual abuse [33], interpersonal violence, stress exposure and stress susceptibility [33] at the micro level, and gender inequality[36, 37] at the macro level, all of which increase the risk for the development of depressive states in women.”

What is the relationship between the mentioned factors and depression associated with agricultural work?

Our response: Thank you for your comment and valuable question, it is our flaw not to narrative clearly about the association between Agricultural labor related depression and female factor. We have already added extended discussion and fully reorganized our logic in the discussion part. To help you better review our revised part, it i as follows:

“The reasons for low life satisfaction among agricultural workers are complex, primarily influenced by the following factors:(1) Economic Pressure[41]: Agricultural workers’ incomes are generally less stable compared to those in non-agricultural occupations, as they are highly susceptible to weather changes, market fluctuations, and other external factors. The seasonal and uncertain nature of agricultural income can impose substantial financial stress, leading to lower life satisfaction.(2) Occupational Health Risks[42]: Agricultural workers face high occupational health risks, including long working hours, exposure to adverse weather conditions, and pesticide exposure mentioned earlier. These health risks negatively impact both physical and mental well-being, reducing overall life s

---

## [Decision Letter · Decision Letter 1]

PONE-D-24-33298R1Can working in agriculture have a favorable effect on depressive symptoms?Life satisfaction as a mediatorPLOS ONE

Dear Dr. Chen,

Thank you for submitting your manuscript to PLOS ONE. After careful consideration, we feel that it has merit but does not fully meet PLOS ONE’s publication criteria as it currently stands. Therefore, we invite you to submit a revised version of the manuscript that addresses the points raised during the review process.

We look forward to receiving your revised manuscript.

Kind regards,

Zahra Lorigooini

Academic Editor

PLOS ONE

**Journal Requirements:**

Reviewers' comments:

Reviewer's Responses to Questions

**Comments to the Author**

1. If the authors have adequately addressed your comments raised in a previous round of review and you feel that this manuscript is now acceptable for publication, you may indicate that here to bypass the “Comments to the Author” section, enter your conflict of interest statement in the “Confidential to Editor” section, and submit your "Accept" recommendation.

Reviewer #1: All comments have been addressed

Reviewer #3: (No Response)

2. Is the manuscript technically sound, and do the data support the conclusions?

Reviewer #1: Yes

Reviewer #3: Partly

3. Has the statistical analysis been performed appropriately and rigorously? 

Reviewer #1: Yes

Reviewer #3: Yes

4. Have the authors made all data underlying the findings in their manuscript fully available?

Reviewer #1: Yes

Reviewer #3: Yes

5. Is the manuscript presented in an intelligible fashion and written in standard English?

Reviewer #1: Yes

Reviewer #3: No

6. Review Comments to the Author

**Reviewer #1: ** Thanks for revising the manuscript based on my earlier comments. The rigor of the manuscript has been enhanced.

**Reviewer #3:**  The literature review is reasonably comprehensive, summarizing relevant studies related to depression, work-related factors, and life satisfaction. However, it could benefit from the incorporation of more recent studies and contrasting findings to provide a wider context. While the identification of gaps in current research is commendable, particularly in relation to job types and depressive symptoms, the paper would be fortified by clearly detailing how the current study directly addresses these identified gaps.

The methodology employed in the study is appropriate for the research questions posed, utilizing a large and representative sample from the China Health and Retirement Longitudinal Study (CHARLS). This approach enhances the generalizability of the findings. The description of the methodology is adequately detailed, covering aspects such as data collection, participant selection, measurement tools, and statistical analyses. However, a more explicit discussion regarding the handling of potential confounders would improve the overall clarity of the methodology section.

The results are presented in a clear and logical manner, with a commendable use of statistics to support the findings. The application of logistic regression and mediation analyses is well-documented. The article makes effective use of tables and figures to present data; however, clearer captions summarizing the key findings for each table and figure would enhance reader comprehension.

The discussion effectively ties back to the research question and objectives, interpreting the results in the context of the existing literature. While the limitations of the study are acknowledged, including its cross-sectional nature and reliance on self-reported measures, a more thorough exploration of the implications of these limitations would enrich the discussion.

The article is generally well-written and clear, although some sections could benefit from reduced jargon and more straightforward language to improve accessibility for a broader audience. The logical structure of the article facilitates a coherent flow from the introduction, through methodology and results, to discussion and conclusion.

The study adheres to ethical standards as evidenced by approval from the Peking University Ethics Committee and informed consent obtained from participants. However, potential conflicts of interest are not explicitly addressed, which should be included to enhance transparency.

The references cited in the article are relevant, although some may be outdated. Incorporating more recent studies could bolster the literature supporting the research. The article appears to follow a consistent citation style; however, a final review is advisable to ensure completeness and accuracy.

Recommendations

To enhance the quality of the article prior to publication, I recommend the following revisions:

• Incorporate more recent studies and contrasting findings to broaden the context and strengthen the literature base.

• Provide a more in-depth discussion of the implications of the study’s limitations, particularly regarding the cross-sectional design and reliance on self-reported measures.

• Include a section addressing potential conflicts of interest to improve transparency and adhere to ethical standards.

• Simplify some sections of the text to reduce jargon and make the article more accessible to a wider audience.

• Add clearer captions that summarize the key findings of each table and figure to enhance reader comprehension.

7. PLOS authors have the option to publish the peer review history of their article (what does this mean? ). If published, this will include your full peer review and any attached files.

**Do you want your identity to be public for this peer review?** For information about this choice, including consent withdrawal, please see our Privacy Policy .

Reviewer #1: **Yes: ** Dickson Adom

Reviewer #3: No

---

## [Author Response · Author response to Decision Letter 2]

2 Apr 2025

Dear Editor

Thank you for your kind comment and rapid arrangement on our revision, we have carefully read and replied to every comment made by 2 reviewers on our manuscript. Please review our revised manuscript and response letter.

Hope everything smooth.

Best Regards,

Yuan Yao Chen

yuanyaochen0401@163.com

Reviewer #1: Thanks for revising the manuscript based on my earlier comments. The rigor of the manuscript has been enhanced.

Dear Reviewer 1,

Thank you very much for your comments. We really appreciate your approval and recognition of our revised work.

Hope everything smooth.

Best Regards,

Yuan Yao Chen

yuanyaochen0401@163.com

Reviewer #3: The literature review is reasonably comprehensive, summarizing relevant studies related to depression, work-related factors, and life satisfaction. However, it could benefit from the incorporation of more recent studies and contrasting findings to provide a wider context. While the identification of gaps in current research is commendable, particularly in relation to job types and depressive symptoms, the paper would be fortified by clearly detailing how the current study directly addresses these identified gaps.

The methodology employed in the study is appropriate for the research questions posed, utilizing a large and representative sample from the China Health and Retirement Longitudinal Study (CHARLS). This approach enhances the generalizability of the findings. The description of the methodology is adequately detailed, covering aspects such as data collection, participant selection, measurement tools, and statistical analyses. However, a more explicit discussion regarding the handling of potential confounders would improve the overall clarity of the methodology section.

The results are presented in a clear and logical manner, with a commendable use of statistics to support the findings. The application of logistic regression and mediation analyses is well-documented. The article makes effective use of tables and figures to present data; however, clearer captions summarizing the key findings for each table and figure would enhance reader comprehension.

The discussion effectively ties back to the research question and objectives, interpreting the results in the context of the existing literature. While the limitations of the study are acknowledged, including its cross-sectional nature and reliance on self-reported measures, a more thorough exploration of the implications of these limitations would enrich the discussion.

The article is generally well-written and clear, although some sections could benefit from reduced jargon and more straightforward language to improve accessibility for a broader audience. The logical structure of the article facilitates a coherent flow from the introduction, through methodology and results, to discussion and conclusion.

The study adheres to ethical standards as evidenced by approval from the Peking University Ethics Committee and informed consent obtained from participants. However, potential conflicts of interest are not explicitly addressed, which should be included to enhance transparency.

The references cited in the article are relevant, although some may be outdated. Incorporating more recent studies could bolster the literature supporting the research. The article appears to follow a consistent citation style; however, a final review is advisable to ensure completeness and accuracy.

Dear Reviewer 3,

Thank you very much for your comments and approval on our manuscript! We have read your comments carefully and made a modification to the original manuscript. The revised part was marked in yellow based on your comments. The following are my responses to your comments.

Recommendations

To enhance the quality of the article prior to publication, I recommend the following revisions:

• Incorporate more recent studies and contrasting findings to broaden the context and strengthen the literature base.

• Provide a more in-depth discussion of the implications of the study’s limitations, particularly regarding the cross-sectional design and reliance on self-reported measures.

• Include a section addressing potential conflicts of interest to improve transparency and adhere to ethical standards.

• Simplify some sections of the text to reduce jargon and make the article more accessible to a wider audience.

• Add clearer captions that summarize the key findings of each table and figure to enhance reader comprehension.

Reply to Reviewer3:

Thank you for your detailed and precious comments on our manuscript, we have carefully read your comments and made revise on the revised manuscript on yellow.

1.For incorporating more recent studies and contrasting findings to broaden the context, we have added new content to support our study background in Introduction Sections:

“While empirical research has demonstrated a significantly elevated prevalence of depressive symptoms among middle-aged and elderly populations residing in rural China compared to their urban counterparts[14]. This epidemiological phenomenon stands in stark contrast to the theoretical postulates of Horticultural Therapy, which advocate the mental health benefits derived from nature immersion. The findings reveal a complex mechanistic relationship between agrarian occupational environments and psychological well-being, indicating that mere proximity to natural elements fails to comprehensively elucidate the etiological pathways through which agricultural labor influences depressive states.”

2.For more in-depth discussion of the implications of the study’s limitations, we have added narration about the cross-sectional design and reliance on self-reported measures in the last part of Discussion Section on yellow.

3.We have complete our conflict of interest narration to improve transparency and adhere to ethical standards, which is as follows:

“The authors declare that they have no known competing financial interests or personal relationships that could have appeared to influence the work reported in this paper.”

4.As for clearer captions, we have adjusted the title of Figure3-5 in order to better summarize the content of the figures. Moreover, we have optimized the narration of Result Sections to more concisely and accurately describe our result.

Best Regards,

Yuan Yao Chen

yuanyaochen0401@163.com

---

## [Decision Letter · Decision Letter 2]

PONE-D-24-33298R2Can working in agriculture have a favorable effect on depressive symptoms?Life satisfaction as a mediatorPLOS ONE

Dear Dr. Chen,

Thank you for submitting your manuscript to PLOS ONE. After careful consideration, we feel that it has merit but does not fully meet PLOS ONE’s publication criteria as it currently stands. Therefore, we invite you to submit a revised version of the manuscript that addresses the points raised during the review process.

We look forward to receiving your revised manuscript.

Kind regards,

Zahra Lorigooini

Academic Editor

PLOS ONE

Journal Requirements:

Reviewers' comments:

Reviewer's Responses to Questions

**Comments to the Author**

1. If the authors have adequately addressed your comments raised in a previous round of review and you feel that this manuscript is now acceptable for publication, you may indicate that here to bypass the “Comments to the Author” section, enter your conflict of interest statement in the “Confidential to Editor” section, and submit your "Accept" recommendation.

Reviewer #3: All comments have been addressed

2. Is the manuscript technically sound, and do the data support the conclusions?

Reviewer #3: Yes

3. Has the statistical analysis been performed appropriately and rigorously? 

Reviewer #3: Yes

4. Have the authors made all data underlying the findings in their manuscript fully available?

Reviewer #3: Yes

5. Is the manuscript presented in an intelligible fashion and written in standard English?

Reviewer #3: Yes

6. Review Comments to the Author

Reviewer #3: • The background is relatively up-to-date, mentioning key studies but could benefit from a broader range of literature around depression and agricultural work.

• The objectives are clearly stated, aiming to explore the association of agricultural work with depressive symptoms and the mediating role of life satisfaction.

• The study methods are sufficiently detailed for replication, including sampling and assessment tools used (CHARLS data, CES-D scale for depression).

• The statistical analyses appear appropriate, utilizing regression methods and sensitivity analyses to ensure robustness in findings.

• The methodology does attempt to address potential biases by including various covariates and controlling for other factors influencing mental health.

• Research was conducted in an ethical manner, approved by the relevant institutional review board.

• The findings are presented clearly, with appropriate statistical results discussed, though incorporating more visual aids could enhance clarity.

• Results align well with stated objectives, showing significant associations between agricultural work, depressive symptoms, and life satisfaction.

• While the initial analyses are thorough, sensitivity analyses and confidence intervals could enhance the validity of the results.

• The authors provide evidence supporting their claims, referencing statistical findings effectively.

• The discussion includes comparisons with existing literature, although a more in-depth exploration of how findings differ from or confirm previous studies could strengthen arguments.

• The alternative interpretations of findings considered are not considered. The authors could expand on alternative interpretations of their findings for a more balanced discussion.

• Conclusions logically follow the data and analyses presented.

• Limitations are acknowledged, but a deeper discussion on their implications could enhance the conclusion.

• The study does suggest avenues for future research, particularly in improving life satisfaction among agricultural workers.

• The manuscript is well-organized, though some sections could benefit from more concise presentations.

• The writing is generally clear and accessible, suitable for readers with a general background in health research.

7. PLOS authors have the option to publish the peer review history of their article (what does this mean? ). If published, this will include your full peer review and any attached files.

**Do you want your identity to be public for this peer review?** For information about this choice, including consent withdrawal, please see our Privacy Policy .

Reviewer #3: No

---

## [Author Response · Author response to Decision Letter 3]

16 Apr 2025

Dear Editor,

Thank you for your kind decision to have a revision chance! We have thoroughly read any comments made by Reviewers and indeed replied. Thank you again!

Best regards,

Yuan Yao Chen

Reviewer #3: 

Dear Reviewer3,

It is our honor to have your approval of our revised manuscript. We really appreciate your precious comments and advice. Thank you again for your highly speak of our work.

• The background is relatively up-to-date, mentioning key studies but could benefit from a broader range of literature around depression and agricultural work.

• The objectives are clearly stated, aiming to explore the association of agricultural work with depressive symptoms and the mediating role of life satisfaction.

• The study methods are sufficiently detailed for replication, including sampling and assessment tools used (CHARLS data, CES-D scale for depression).

• The statistical analyses appear appropriate, utilizing regression methods and sensitivity analyses to ensure robustness in findings.

• The methodology does attempt to address potential biases by including various covariates and controlling for other factors influencing mental health.

• Research was conducted in an ethical manner, approved by the relevant institutional review board.

• The findings are presented clearly, with appropriate statistical results discussed, though incorporating more visual aids could enhance clarity.

• Results align well with stated objectives, showing significant associations between agricultural work, depressive symptoms, and life satisfaction.

• While the initial analyses are thorough, sensitivity analyses and confidence intervals could enhance the validity of the results.

• The authors provide evidence supporting their claims, referencing statistical findings effectively.

• The discussion includes comparisons with existing literature, although a more in-depth exploration of how findings differ from or confirm previous studies could strengthen arguments.

• The alternative interpretations of findings considered are not considered. The authors could expand on alternative interpretations of their findings for a more balanced discussion.

• Conclusions logically follow the data and analyses presented.

• Limitations are acknowledged, but a deeper discussion on their implications could enhance the conclusion.

• The study does suggest avenues for future research, particularly in improving life satisfaction among agricultural workers.

• The manuscript is well-organized, though some sections could benefit from more concise presentations.

• The writing is generally clear and accessible, suitable for readers with a general background in health research.

Reply: It is our honor to have your approval of our revised manuscript. We really appreciate your precious comments and advice. And we would continue our further study on Job diversities with mental health and try to figure out more reasonable explanation based on our current research, especially try to decline our limitations during collecting data and designing our study.

Thank you again for your highly speak of our work.

Best regards,

Yuan Yao Chen

---

## [Decision Letter · Decision Letter 3]

PONE-D-24-33298R3Can working in agriculture have a favorable effect on depressive symptoms?Life satisfaction as a mediatorPLOS ONE

Dear Dr. Chen,

Thank you for submitting your manuscript to PLOS ONE. After careful consideration, we feel that it has merit but does not fully meet PLOS ONE’s publication criteria as it currently stands. Therefore, we invite you to submit a revised version of the manuscript that addresses the points raised during the review process.

We look forward to receiving your revised manuscript.

Kind regards,

Zahra Lorigooini

Academic Editor

PLOS ONE

Journal Requirements:

Reviewers' comments:

Reviewer's Responses to Questions

**Comments to the Author**

1. If the authors have adequately addressed your comments raised in a previous round of review and you feel that this manuscript is now acceptable for publication, you may indicate that here to bypass the “Comments to the Author” section, enter your conflict of interest statement in the “Confidential to Editor” section, and submit your "Accept" recommendation.

Reviewer #3: All comments have been addressed

2. Is the manuscript technically sound, and do the data support the conclusions?

Reviewer #3: Partly

3. Has the statistical analysis been performed appropriately and rigorously? 

Reviewer #3: Yes

4. Have the authors made all data underlying the findings in their manuscript fully available?

Reviewer #3: Yes

5. Is the manuscript presented in an intelligible fashion and written in standard English?

Reviewer #3: Yes

6. Review Comments to the Author

Reviewer #3: The study addresses a critical issue and Utilizing the CHARLS 2015 dataset (n=6,856) enhances statistical power and generalizability to China’s middle-aged and elderly population. The use of CES-D for depressive symptoms and SWLS for life satisfaction strengthens the reliability of the outcomes. Inclusion of demographic, behavioral, and social factors (e.g., age, gender, social activity) reduces confounding bias. Mediation analysis via Baron and Kenny’s approach and Sobel test appropriately tests the hypothesized pathways.

However, consider the following concern to address them Limits causal inference. The observed associations could reflect reverse causality (e.g., depressive individuals may self-select into agricultural work).

1. Excluding 11,270 participants due to missing values risks selection bias. Multiple imputation or sensitivity analyses are needed to validate findings.

2. Dichotomizing jobs into "agricultural" vs. "non-agricultural" ignores heterogeneity within agricultural work (e.g., subsistence vs. commercial farming). Clarify contradictions (e.g., job type’s non-significance in logistic regression vs. abstract’s emphasis on self-employed risk).

3. Reliance on one question ("How satisfied are you with your life?") may lack depth compared to multi-item scales like SWLS.

4. Life satisfaction mediated only 5.13% of the total effect, questioning its practical significance. Other mediators (e.g., income stability, social support) were overlooked.

5. Self-employed agricultural workers showed more protective factors, conflicting with literature on self-employment stressors. The rationale for this is underexplored.

6. The 2015 dataset may not reflect recent socioeconomic changes in rural China (e.g., digitalization, policy shifts).

7. Inconsistent table/figure numbering, (e.g., Fig. 3–7 not provided), and unclear reporting of propensity score matching methodology. Provide full details on propensity score matching (variables, method, balance diagnostics).

Highlight them in limitation and recommend them for future consideration.

1. Use longitudinal data to establish temporal relationships.

2. Apply multiple imputation for missing data and report sensitivity analyses.

3. Refine job type categorization (e.g., differentiate farming practices, land ownership).

4. Consider alternative regression models (e.g., Poisson) for common outcomes to avoid overestimating odds ratios.

5. Explore additional mediators (e.g., economic stability, pesticide exposure) to explain the weak mediation effect of life satisfaction.

6. Include all figures/tables referenced (e.g., correlation matrix, mediation model).

7. Address any ethical and temporal relevance related to mental health.

7. PLOS authors have the option to publish the peer review history of their article (what does this mean? ). If published, this will include your full peer review and any attached files.

**Do you want your identity to be public for this peer review?** For information about this choice, including consent withdrawal, please see our Privacy Policy .

Reviewer #3: No

---

## [Author Response · Author response to Decision Letter 4]

23 Apr 2025

Dear Reviewer3,

Thank you for your detailed comments and sincere help to help us improve our further research quality and enhance our narration of our limitations. We are sorry that due to the structure our original data set and observational study, it is hard to make deeper revision on the current study, however, we would continue to make our future research more complete under your guidance.

The study addresses a critical issue and Utilizing the CHARLS 2015 dataset (n=6,856) enhances statistical power and generalizability to China’s middle-aged and elderly population. The use of CES-D for depressive symptoms and SWLS for life satisfaction strengthens the reliability of the outcomes. Inclusion of demographic, behavioral, and social factors (e.g., age, gender, social activity) reduces confounding bias. Mediation analysis via Baron and Kenny’s approach and Sobel test appropriately tests the hypothesized pathways.

However, consider the following concern to address them Limits causal inference. The observed associations could reflect reverse causality (e.g., depressive individuals may self-select into agricultural work).

1. Excluding 11,270 participants due to missing values risks selection bias. Multiple imputation or sensitivity analyses are needed to validate findings.

2. Dichotomizing jobs into "agricultural" vs. "non-agricultural" ignores heterogeneity within agricultural work (e.g., subsistence vs. commercial farming). Clarify contradictions (e.g., job type’s non-significance in logistic regression vs. abstract’s emphasis on self-employed risk).

3. Reliance on one question ("How satisfied are you with your life?") may lack depth compared to multi-item scales like SWLS.

4. Life satisfaction mediated only 5.13% of the total effect, questioning its practical significance. Other mediators (e.g., income stability, social support) were overlooked.

5. Self-employed agricultural workers showed more protective factors, conflicting with literature on self-employment stressors. The rationale for this is underexplored.

6. The 2015 dataset may not reflect recent socioeconomic changes in rural China (e.g., digitalization, policy shifts).

7. Inconsistent table/figure numbering, (e.g., Fig. 3–7 not provided), and unclear reporting of propensity score matching methodology. Provide full details on propensity score matching (variables, method, balance diagnostics).

Highlight them in limitation and recommend them for future consideration.

1. Use longitudinal data to establish temporal relationships.

2. Apply multiple imputation for missing data and report sensitivity analyses.

3. Refine job type categorization (e.g., differentiate farming practices, land ownership).

4. Consider alternative regression models (e.g., Poisson) for common outcomes to avoid overestimating odds ratios.

5. Explore additional mediators (e.g., economic stability, pesticide exposure) to explain the weak mediation effect of life satisfaction.

6. Include all figures/tables referenced (e.g., correlation matrix, mediation model).

7. Address any ethical and temporal relevance related to mental health.

Authors’ Reply: Thank you for your kind comment! We have added several advice and comment into our Section Discussion and Limitations, which were highlighted in yellow in our revised manuscript. Moreover, we have checked and revised the miss cited table&figure in the manuscript, cited them in the right order and reapplied them into the system. The revised limitations were as follows:

“This study has several limitations. First, it is based on cross-sectional data from the 2015 China Health and Retirement Longitudinal Study (CHARLS), which limits the ability to establish a clear temporal sequence between agricultural employment and depressive symptoms. Consequently, reverse causality cannot be ruled out; for instance, individuals with depression may be more inclined to engage in agricultural work.Longitudinal designs are essential for understanding the temporal relationship between job type and mental health outcomes. Existing longitudinal investigations have illuminated that the depressive symptoms is not static; rather, it can fluctuate over time, which cross-sectional studies may not adequately represent. Longitudinal research highlights the dynamic relationship between job type and mental health, with studies showing that unstable employment can lead to increased depressive symptoms over time[77] and work-related stressors can exacerbate mental health issues[78]. Future research should employ longitudinal data to more accurately elucidate causal relationships between variables. Second, a substantial number of samples (n = 11,270) were excluded due to missing covariate data, which may introduce selection bias. Future studies should consider employing multiple imputation techniques to handle missing data and conduct sensitivity analyses to assess the robustness of the findings. Additionally, the classification of occupational types was simplistic, dividing them merely into "agricultural" and "non-agricultural," failing to capture the heterogeneity within agricultural work, such as subsistence versus commercial farming and land ownership differences. Future research should refine occupational classifications to more comprehensively understand the impact of occupational types on depressive symptoms. The study also employed logistic regression to analyze depressive symptoms, a common outcome variable, which may overestimate odds ratios. Future studies might consider alternative regression models, such as Poisson regression, to obtain more accurate estimates. Although this study found that life satisfaction had a mediating effect between agricultural work and depressive symptoms, the effect was weak, accounting for only 5.13% of the total effect. This may be due to the omission of other potential mediating variables, such as economic stability and pesticide exposure. Future research should explore additional mediators to more fully understand the mechanisms by which agricultural work influences depressive symptoms. Finally, the 2015 data used in this study may not capture recent socioeconomic changes in rural China, such as digital development and policy shifts. Additionally, mental health research involves sensitive information, necessitating careful ethical considerations to ensure participant privacy and informed consent during data collection and analysis. Future studies should focus on the temporal relevance of data and enhance ethical review and participant protection measures.”

Best regards,

Yuan Yao Chen

---

## [Editor Report · Decision Letter 4]

PONE-D-24-33298R4Can working in agriculture have a favorable effect on depressive symptoms?Life satisfaction as a mediatorPLOS ONE

Dear Dr. Chen,

Thank you for submitting your manuscript to PLOS ONE. After careful consideration, we feel that it has merit but does not fully meet PLOS ONE’s publication criteria as it currently stands. Therefore, we invite you to submit a revised version of the manuscript that addresses the points raised during the review process.

**To ensure clarity and thoroughness in addressing their comments, we kindly request that you respond to each reviewer’s comment separately in your response file, rather than addressing all comments collectively.**

We look forward to receiving your revised manuscript.

Kind regards,

Zahra Lorigooini

Academic Editor

PLOS ONE
---

## [Author Response · Author response to Decision Letter 5]

7 May 2025

Dear Editor,

Thank you for your kind reminder, we have searched and checked all the cited references, and made sure that all the list is complete, correct and no paper were retracted in present. Thank you for the kind reminder again!

Best Regards,

Yuan Yao Chen

---

## [Editor Report · Decision Letter 5]

PONE-D-24-33298R5Can working in agriculture have a favorable effect on depressive symptoms?Life satisfaction as a mediatorPLOS ONE

Dear Dr. Chen,

Thank you for submitting your manuscript to PLOS ONE. After careful consideration, we feel that it has merit but does not fully meet PLOS ONE’s publication criteria as it currently stands. Therefore, we invite you to submit a revised version of the manuscript that addresses the points raised during the review process.

**ACADEMIC EDITOR:**

I have carefully reviewed the manuscript and would like to share some editorial suggestions to enhance the clarity and professionalism of the text:

**Inconsistent Spacing:**In several instances, there are missing spaces after periods and commas, e.g., "depressive symptoms.Life satisfaction" should be "depressive symptoms. Life satisfaction."Extra spaces were found between words in sentences like "working in agriculture," which should be corrected to "working in agriculture."In many instances, reference numbers are directly attached to the last word, e.g., "study shows that depression is prevalent among older adults[12]" should be "study shows that depression is prevalent among older adults [12]."
**Improper Hyphenation and Spacing:**Some compound terms such as "self-employedagricultural workers" lack appropriate spacing. This should be revised to "self-employed agricultural workers."**Extra Periods:**In certain parts of the manuscript, extra periods were identified, e.g., "data were analyzed.." This should be corrected to "data were analyzed."
**Spacing Around Statistical Values:**In statistical reporting, there should be a space before and after inequality symbols, e.g., "p<0.001" should be "p < 0.001."
**Inconsistent Use of Punctuation:**Some sections contain inconsistent use of commas and semicolons. Please ensure that list items are separated correctly and that sentences are punctuated properly.
**Suggestions for Improvement:**I recommend a thorough proofreading to ensure consistency in spacing, punctuation, and formatting.Utilizing tools such as Grammarly or the built-in proofing tools in Microsoft Word may assist in identifying minor errors.

**Note:** I have previously mentioned that the manuscript should be edited according to the PLOS ONE formatting guidelines. However, the issues mentioned above are still present in the text. I recommend careful alignment with the PLOS ONE standards to avoid further revisions.

I hope these suggestions help in enhancing the quality and readability of the manuscript. If you require further clarification or assistance with revisions, please feel free to reach out.

We look forward to receiving your revised manuscript.

Kind regards,

Zahra Lorigooini

Academic Editor

PLOS ONE
---

## [Author Response · Author response to Decision Letter 6]

19 May 2025

Dear Editor,

We sincerely appreciate the editor’s meticulous review and constructive feedback. Below is our point-by-point responsethe issues raised, along with the revisions made to the manuscript:

I have carefully reviewed the manuscript and would like to share some editorial suggestions to enhance the clarity and professionalism of the text:

Inconsistent Spacing:

In several instances, there are missing spaces after periods and commas, e.g., "depressive symptoms.Life satisfaction" should be "depressive symptoms. Life satisfaction."

Authors’ reply Thank you for your thoroughly check and review, all instances of missing or extra spacing have been corrected. For example, revised“depressive symptoms.Life satisfaction”to“depressive symptoms. Life satisfaction.” Adjusted spacing in phrases like “working in agriculture,” to ensure proper formatting, which were all highlighted in yellow.

Extra spaces were found between words in sentences like "working in agriculture," which should be corrected to "working in agriculture."

In many instances, reference numbers are directly attached to the last word, e.g., "study shows that depression is prevalent among older adults[12]" should be "study shows that depression is prevalent among older adults [12]."

Authors’ reply Thank you for your thoroughly check and review, spaces have been added between text and reference numbers in all instances. We have made sure all reference numbers with spaces attached to the last word.

Improper Hyphenation and Spacing:

Some compound terms such as "self-employedagricultural workers" lack appropriate spacing. This should be revised to "self-employed agricultural workers."

Authors’ reply Thank you for your thoroughly check and review, all compound terms have been standardized with appropriate spacing: Corrected “self-employedagricultural workers” to “self-employed agricultural workers.”The corrections were in yellow in the revised manuscript.

Extra Periods:

In certain parts of the manuscript, extra periods were identified, e.g., "data were analyzed.." This should be corrected to "data were analyzed."

Spacing Around Statistical Values:

In statistical reporting, there should be a space before and after inequality symbols, e.g., "p<0.001" should be "p < 0.001."

Authors’ reply Thank you for your thoroughly check and review, redundant periods have been removed. Examples include: Revised “data were analyzed..” to “data were analyzed.”Checked for duplicated punctuation in tables, footnotes, and body text.

Inconsistent Use of Punctuation:

Some sections contain inconsistent use of commas and semicolons. Please ensure that list items are separated correctly and that sentences are punctuated properly.

Suggestions for Improvement:

I recommend a thorough proofreading to ensure consistency in spacing, punctuation, and formatting.

Utilizing tools such as Grammarly or the built-in proofing tools in Microsoft Word may assist in identifying minor errors.

Authors’ reply Thank you for your thoroughly check and review, punctuation has been standardized: Corrected list separators (e.g., revised “factors such as work pressure [5], work hours [6, 7]” to ensure commas and semicolons align with journal guidelines). Adjusted sentence structures for clarity and consistency (e.g., removed misplaced semicolons in tables). Moreover, we’ve check our grammar and revised some sentences details.

Best regards,

Yuan Yao Chen

---

## [Editor Report · Decision Letter 6]

Can working in agriculture have a favorable effect on depressive symptoms?Life satisfaction as a mediator

PONE-D-24-33298R6

Dear Dr. Chen,

We’re pleased to inform you that your manuscript has been judged scientifically suitable for publication and will be formally accepted for publication once it meets all outstanding technical requirements.

Kind regards,

Zahra Lorigooini

Academic Editor

PLOS ONE
---

## [Editor Report · Acceptance letter]

PONE-D-24-33298R6

PLOS ONE

Dear Dr. Chen,

I'm pleased to inform you that your manuscript has been deemed suitable for publication in PLOS ONE. Congratulations! Your manuscript is now being handed over to our production team.

Kind regards,

on behalf of

Prof. Zahra Lorigooini

Academic Editor

PLOS ONE